# Brain-Like Processing Pathways Form in Models With Heterogeneous Experts

**Jack Cook**[1]    **Danyal Akarca**[2]    **Rui Ponte Costa**[1,*]    **Jascha Achterberg**[1,*]

[1]Centre for Neural Circuits and Behaviour, University of Oxford
[2]Department of Electrical and Electronic Engineering, Imperial College London
[*]Joint senior authors

## Abstract

The brain is made up of a vast set of heterogeneous regions that dynamically organize into pathways as a function of task demands. Examples of such pathways can be found in the interactions between cortical and subcortical networks during learning, or in sub-networks specializing for task characteristics such as difficulty or modality. Despite the large role these pathways play in cognition, the mechanisms through which brain regions organize into pathways remain unclear. In this work, we use an extension of the Heterogeneous Mixture-of-Experts architecture to show that heterogeneous regions do not form processing pathways by themselves, implying that the brain likely implements specific constraints which result in the reliable formation of pathways. We identify three biologically relevant inductive biases that encourage pathway formation: a routing cost imposed on the use of more complex regions, a scaling factor that reduces this cost when task performance is low, and randomized expert dropout. When comparing our resulting *Mixture-of-Pathways* model with the brain, we observe that the artificial pathways in our model match how the brain uses cortical and subcortical systems to learn and solve tasks of varying difficulty. In summary, we introduce a novel framework for investigating how the brain forms task-specific pathways through inductive biases, and the effects these biases have on the behavior of Mixture-of-Experts models.

## 1   Introduction

The brain is made up of many heterogeneous regions distinguished by features such as connectivity, cell types, neurotransmitters, and functional specialization [1–4]. To support complex behavior, the mammalian brain dynamically organizes these regions into diverse networks and processing pathways [5], allowing it to adapt to different inputs and task demands. This principle spans sensory systems [6–8], cognitive networks [9], emotion-related circuits [10], and face perception [11]. Notably, pathway formation is highly dynamic: regions can participate in many pathways, allowing cognitive processes to arise from the joint activations of specific groups of regions. While theoretical work has shown how heterogeneous regions and modules can develop within networks [12–14], how these organize into large-scale pathways remains poorly understood.

The importance of studying pathway formation and coordination extends beyond neuroscience: it is also becoming increasingly relevant in machine learning research. As models have evolved from small networks with a couple of layers to large system-level architectures, achieving complex function while maintaining efficiency has become critical. One recent development toward this goal is the Mixture-of-Experts (MoE) architecture [15, 16], which contains specialized experts that are selectively activated based on the current input. This should create pathways between experts that selectively respond to inputs of varying complexity [17] to make efficient use of computational resources [18, 19]. However, this theorized specialization of experts appears to be limited in practice [20, 21], making it difficult for specialized task-complexity-related pathways to form.

39th Conference on Neural Information Processing Systems (NeurIPS 2025).

These findings raise the question, how do stable and functionally relevant pathways form in networks of distributed heterogeneous experts? Do heterogeneous regions automatically group into such pathways, or are additional priors required? Once pathways develop in models, do they show the same context-aware adaptability that has been observed in the brain? To address these questions, we introduce a neural network architecture made up of heterogeneous experts to study the conditions under which processing pathways form, and the degree to which these pathways resemble those studied in the brain. Specifically, we adapt the Heterogeneous Mixture-of-Experts (HMoE) architecture [22, 23], in which information may be dynamically routed to computational experts, or regions, of varying sizes. In our model, unlike prior work, each expert is implemented as a recurrent network that could be considered a standalone model or brain region. We study the pathway formation in this architecture while we train models to learn 82 time-series-based cognitive tasks of varying difficulty [24]. Through these analyses, we find:

- Layers of heterogeneous experts do not automatically form recognizable pathways.
- Instead, inductive biases are required for pathways to form: (i) a routing cost that penalizes the model for using larger experts, (ii) scaling the routing cost based on task performance, and (iii) random expert dropout. These result in the formation of a *Mixture-of-Pathways*.
- The pathways that form in our new *Mixture-of-Pathways* architecture mirror the interactions between cortical and subcortical pathways in the brain during learning, and are in line with the dynamics of the brain's multiple-demand system [9, 25].

To arrive at these findings, in the following we start by analyzing the usage of experts of a baseline model with HMoE layers. We then develop specific inductive biases that encourage pathways to form, before finally comparing these pathways to observations made in the brain.

## 2  Related Work

Brain-like modularity and regional heterogeneity can be induced in neural networks through priors and training procedures to explain how such features develop in the brain [14, 26–30]. The priors that are especially relevant in the context of this work relate to metabolic cost and energy efficiency, which are crucial in determining the brain's circuitry and function [13, 31–34].

While the above work often focuses on starting from fully-connected networks to observe the formation of modules and regions, modeling multi-region interactions has also become possible with modern methods [35–40]. This line of work has revealed how joint computation can be implemented through interactions between independent modules [5]. However, these multi-region models generally predefine a specific circuit structure with a small set of regions, preventing further study on how regions come to interact in the first place. Notable work that allows for dynamic (non-fixed) interaction of multiple independent regions often assumes networks which are not able to learn tasks [41, 42], though [43] stands out with a trainable network made up of individual RNNs. Their multi-region networks can change their connectivity during learning, but cannot route information based on task context. The work most closely aligned with our goal is [44], which studies how spatial (metabolic) constraints in feed-forward networks can form visual processing streams. However, it too does not consider how regions may change their interaction as a function of context, and does not allow for solving standard time-series-based cognitive tasks.

In the context of artificial intelligence, the introduction outlined how the popular Mixture-of-Experts architecture [15, 16] is relevant to our question of how specialized regions dynamically organize into processing pathways. Recent efforts to build networks out of experts that vary in terms of their architecture [17, 22, 23] and function [20] are especially relevant here. Moreover, work has argued that routing pathways are a powerful method for handling the complex data-flow of these otherwise efficient architectures [19, 45], but without investigating how adaptable pathways can be encouraged to form. In neuromorphic computing it has been shown to be possible to implement brain-like visual processing pathways to achieve efficient processing [46], but with a predefined static architecture.

## 3  Methods

In this work, we aim to identify the mechanisms by which pathways form between heterogeneous regions, and how those pathways are used across a diverse set of tasks. To study this computationally,

we need a model made up of heterogeneous experts, analogous to brain regions, that work together to solve many different tasks. We create such a model by extending the Heterogeneous Mixture-of-Experts architecture [22] and training it on the Mod-Cog set of time-series-based cognitive tasks [24].

## 3.1 Model Architecture: Heterogeneous Mixture-of-Experts

Mixture-of-Experts models (MoEs) [15, 16] are characterized by their layers, which contain multiple smaller models alongside a router model, which decides which experts should process the input at each timestep. Specifically, the router determines the weight with which each expert contributes to the layer's final output. Experts can also be excluded, by setting an expert's weight to zero. In most modern MoEs, the experts are large scale MLPs placed in between attention layers, which are typically activated at every timestep. In Heterogeneous Mixture-of-Expert models [22] (HMoEs), the experts can vary in terms of their sizes and activation functions. We extend the HMoE architecture with several significant adaptations. Namely, each layer of our model contains three experts: two GRUs with 16 and 32 neurons respectively, and one skip connection, which allows the model to choose to perform no computation for a given timestep [17]. Our implementation uses GRUs with 64 neurons as routers and does not include any additional layers between HMoE layers. We use this setup as a baseline for our investigations

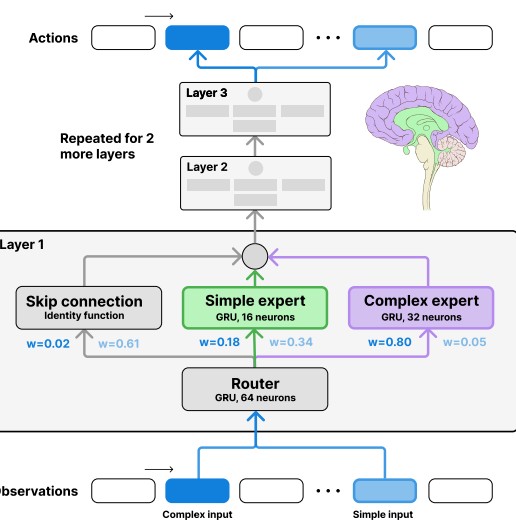

Figure 1: **Schematic of our baseline model architecture**. Information is passed through three layers, each of which can dynamically route information to experts of different computational complexity.

(Figure 1). In the following sections, we will introduce additional inductive biases to this baseline architecture, resulting in our final Mixture-of-Pathways model. The algorithm for this full model is described in Algorithm 1.

---

**Algorithm 1:** Mixture-of-Pathways training protocol. Full details in Appendix A.1.

---

**Data:** Task dataset $D$, Experts $E = \{e_1, e_2, \ldots, e_n\}$, Routers $R = \{r_1, r_2, \ldots, r_n\}$
Initialize routers and heterogeneous experts, set $h_0$ to task input;
**for** *each training step* **do**
    Sample batch $b$ from $D$;
    **for** *each task $i$, timestep $t \in b$* **do**
        **for** *each layer $l$ with experts $e_j \in l$* **do**
            Compute routing weights $w_j = \text{softmax}(r_l(h_{l-1}))$ ;
            Apply expert dropout (Not in baseline architecture; see Section 4.2) ;
            Compute expert activations $w_j$ for each expert $e_j \in l$ ;
            Combine outputs: $h_l = \sum_j w_j h_{l,j}$;
        **end**
        Compute baseline model losses: $L_{\text{fix}}$ and $L_{\text{response},i}$ ;
        Compute $L_{\text{routing}}$ loss (Not in baseline architecture; see Section 4.1) ;
        Compute $L_{\text{total}} = L_{\text{fix}} + L_{\text{routing}} + \sum_i L_{\text{response},i}$;
    **end**
    Update parameters using Schedule-Free AdamW optimizer [47];
**end**

---

## 3.2 Evaluation with Cognitive Tasks

To evaluate how pathways are formed and used across tasks with different characteristics, we use the Mod-Cog task set, which contains 82 time-series-based cognitive tasks [24]. This is an expansion of

the popular NeuroGym framework [48], which contains tasks like Go-NoGo or two-stimuli integration tasks (see Appendix A.2 for task details). Importantly for us, the tasks vary in difficulty due to their varied inputs, decision rules, and delay lengths. Generally, tasks range from 0.8 to 3 seconds in duration, during which we sample information at timesteps 100 milliseconds apart. At each timestep, the model receives a 115-dimensional input made up of four components: a 1-dimensional fixation input, two 16-dimensional stimuli, and an 82-dimensional one-hot encoding of the active task, which we pass through a 16-dimensional learned embedding layer. While the fixation input is active, the model should always output zero. After the fixation period, the model needs to use the observed information to output the correct choices during the response period.

We train our models over 10 epochs, each containing 1000 training steps. At each training step, models are given a $128 \times 350 \times 115$ matrix of input data, representing 128 batches of task sequences that are 350 timesteps long, with 115 features at each timestep. These task sequences contain many individual tasks: the average task is about 20 timesteps long, meaning that in each batch, models observe about 27 trials of each task. Models are trained with a cross entropy loss $L_{\text{response},i}$ for the correct response during the response period of task $i$. An additional fixation loss $L_{\text{fix}}$ encourages the model to output zero during the fixation period. All losses used in our analyses are detailed in Appendix A.3. Training one model takes roughly 1 hour on a single NVIDIA T4 GPU. Details on implementation and code access are outlined in Appendix A.1.

Once models have learned to solve each task by routing information between experts, we can study the conditions under which processing pathways form between layers. In the following we will first study the routing behavior of our baseline architecture. We will then show how the additional inductive biases described in Algorithm 1 result in the formation of pathways. Finally, we will test the degree to which these pathways resemble established processing pathways in the brain.

## 4 What Causes Pathways to Form?

In this section, we investigate the conditions under which pathways form between layers of heterogeneous experts. We set three criteria to determine whether pathways have formed:

1. Pathways should be **consistent** with respect to tasks, meaning that when two models are trained to solve the same tasks, they should have structurally similar pathways.
2. Pathways should be **self-sufficient**, meaning that when experts outside of a pathway are removed, then the model's overall performance should remain largely intact.
3. Pathways should be **distinct**, meaning that several different pathways should be used to solve groups of tasks with varying characteristics.

### 4.1 Pathway Consistency

To see if experts form consistent, task-driven pathways, we train 20 randomly initialized models with the same settings and compare their routing patterns on the same set of 82 cognitive tasks. This allows us to examine whether models use similar sets of experts to solve the same tasks, such as whether smaller experts are reliably used to solve simpler tasks, or vice versa. We first do this with a baseline model made up of three HMoE layers, and then test each model on 50 trials of each task while recording the routing weights assigned to each expert at each timestep ($w$ values in Figure 1). To test whether routing is stable across training runs, we use these weights to calculate each model's *Learned Pathway Complexity* for each task $i$ ($LPC_i$) as follows:

$$LPC_i = \frac{1}{T_i} \sum_{t}^{T_i} \sum_{j}^{E} w_{i,j,t} s_j^2 \tag{1}$$

This metric is calculated by multiplying the weight $w_{i,j,t}$ assigned by the router to each expert $j$ at each timestep $t$ by the squared size $s_j^2$ of expert $j$ while the model solves task $i$. This is then averaged across the total timesteps $T_i$ of each task $i$ to ensure that longer tasks are not biased toward having larger $LPC$s. This results in a $LPC$ value for each of the 82 tasks and 20 model runs (see Appendix A.4 for an example calculation). The squaring of expert sizes is motivated by the $O(s_j^2)$ cost of storing each expert's weight matrix in memory, and we expand on the suitability of using $s_j$ as

a measure of each expert's complexity in Appendix A.5. Skip connections are free to use. To measure pathway consistency, we can now correlate this list of $LPC$ values across training runs. For the baseline model, we find that models are not consistent across training runs (mean pairwise correlation of 0.0324, Figure 2), suggesting that the baseline model does not form any stable and task-related processing pathways by default. Therefore, we next want to explore which specific inductive biases may result in such pathways.

Theories of metabolic optimization and cost minimization are core parts of our understanding of the brain's computations [33, 49, 50]. The reduction of energy consumption has been a powerful source of priors for building brain-like neural networks [13, 31, 34] and more generally achieving brain-inspired computing [46, 51]. Hence we hypothesize that regularizing the routing weights by making it more expensive to route to more complex experts might cause replicable pathways to develop, as observed in the brain. We do so by incorporating the $LPC_i$ (from Equation 1) into the model's loss, making it more costly for the model to activate more complex experts. Finally, to avoid convergence on the local minimum of only using the smallest experts without solving any tasks[1], we add a normalization strategy, dividing each $LPC_i$ by $L_{\text{response},i}$, the cross-entropy loss measuring the model's performance on task $i \in \mathcal{T}$. In addition to helping with convergence, this normalization term can also be viewed as helping our model more directly control the cognitive effort, or processing power, with which it solves a task. We discuss this further in Section 6.

Adding these additional components to the loss results in the following equation, where $L_{\text{fix}}$ and $L_{\text{response},i}$ are the standard task-based losses described in Section 3.2. A small value $\epsilon$ is added to ensure that if the model solves tasks perfectly, the routing loss is not $\infty$. All loss calculations are outlined in detail in Appendix A.3 and A.4.

$$L = L_{\text{fix}} + \sum_{i}^{\mathcal{T}} (L_{\text{response},i} + \frac{\alpha LPC_i}{L_{\text{response},i} + \epsilon}) \tag{2}$$

We now evaluate the routing consistency of models trained with this custom loss function. Figure 2 shows that our expectations are confirmed: on its own, adding the $LPC_i$ for each task creates more consistent routing (mean pairwise correlation of 0.15, significant over baseline with $p < 0.01$). Scaling this term by the model's performance on each task $L_{\text{response},i}$ amplifies this effect, encouraging models across training runs to converge on more consistent routing patterns (mean pairwise correlation of 0.71, significant over baseline with $p < 0.001$).

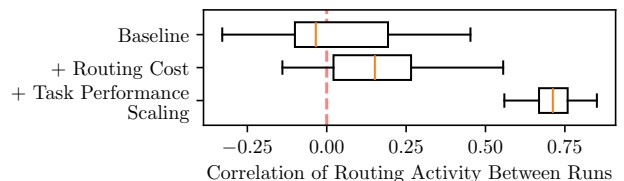

Figure 2: **Models trained with routing cost and task-based scaling exhibit more stable routing.** Correlations are calculated between the routing patterns across 20 training runs of each model setup.

## 4.2 Self-Sufficiency of Pathways

Our second criterion measures whether formed pathways are self-sufficient, meaning that removing an expert that is not part of the currently activated pathway should only minimally impact the performance of the model. To test for self-sufficiency, we first evaluate whether models are still able to perform tasks when they are prevented from using experts that have been assigned low routing weights. We find that our baseline models are extremely sensitive to this deactivation: if they are prevented from using experts with $w < 0.025$, which only contribute 2.5% or less to each layer's output, average task accuracy drops from 98.2% to 16.4%. This shows that while models learn replicable routing patterns, these are not yet pathways, as they rely on all the experts.

We speculate that pathway self-sufficiency can be achieved by stochastic dropout of experts with low routing weights. Dropout is especially interesting as it is an established principle for achieving more

---

[1]Note that routers converging to local minima is an established phenomenon in Mixture-of-Experts models, as there is a bias to rely on the expert that learns the task, or decreases the overall loss as in our case, first [16, 22, 52, 53].

robust neural networks [54] and has also been linked to the stochastic nature of signal processing in neuroscience [55, 56]. We implement *expert dropout* by randomly deactivating experts that contribute very little to the output during training. The probability $p_j$ with which expert $j$ is deactivated at a given timestep is determined as follows:

$$p_j = \begin{cases} \beta - \frac{\beta}{\gamma} w_j, & \text{if } w_j < \gamma \\ 0, & \text{otherwise} \end{cases} \tag{3}$$

We set $\gamma$ to 0.1, meaning that experts contributing 10% or more to the output of a layer are never deactivated. As this contribution weight decreases to zero, this probability increases linearly to $\beta$. To identify how much dropout is needed to improve robustness, we train 11 groups of models with $\beta$ values ranging from $0, 0.1, ..., 0.9, 1$ using Equation 3, with 10 models in each group. We evaluate each model on 50 trials per task, blocking experts with routing weights below 11 values: $0, 0.025, \ldots, 0.225, 0.25$.

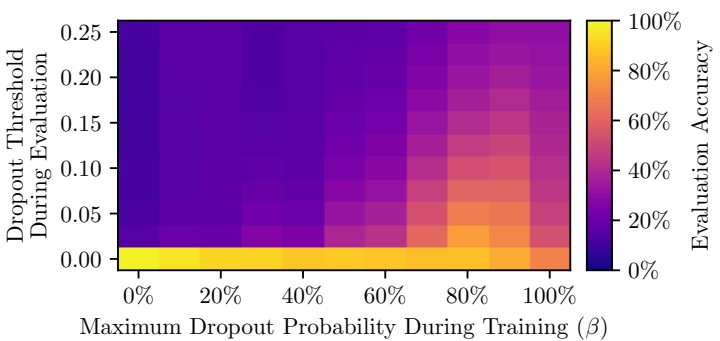

Figure 3: **Model accuracy after removing low-weighted experts across different training dropout levels.** If models trained without dropout ($\beta = 0$) are prevented from using experts that contribute very little to the output, accuracy drops precipitously, from 98.2% to 15.1%. By comparison, the accuracy of models trained with a maximum dropout value of $\beta = 0.8$ only drops from 86.5% to 77.7%.

Accuracy is averaged across the 10 models in each group. We find that expert dropout has a relatively minor impact on performance, while significantly improving the robustness of the pathways that form. Figure 3 shows that for models trained with a maximum dropout of 80% ($\beta = 0.8$), this drop in accuracy is small (from 85.8% to 74.4%). This motivates us to set $\beta = 0.8$ for our models in the remainder of this work. Note that routing consistency (Section 4.1) remains high with dropout, shown through an average pairwise correlation of 0.51, which is significant over the baseline with $p < 0.0001$ (see Appendix A.4).

## 4.3 Distinct Pathways Across and Within Tasks

For our final criterion, we want to identify whether meaningful patterns of expert usage develop across tasks and timescales within our model. To do this, we record the routing patterns across 50 trials of each task for both the baseline model and our final model, which is trained using the routing cost with task performance scaling and expert dropout. To visualize how routing varies across layers, tasks, and time, we average routing patterns for each task in three phases: (i) before the stimulus is shown, (ii) while the stimulus is shown, and (iii) during the response period. We apply K-means clustering ($k = 10$) to these matrices to identify groups of tasks that use similar pathways.

In our model, we observe a structured usage of expert pathways (Figure 4): during the pre-stimulus phase, models primarily rely on the cheap skip connections, as no information needs to be processed yet. For some tasks, increasingly complex experts are activated with the onset of stimuli. However, since the model still only needs to output zero during this phase, most tasks continue to leverage the cheap 'all-skip' pathway until the response phase. During this final phase, we see very rich dynamics of pathways being differentially activated across tasks and time periods in our model. This shows how processing pathways interact over the time of a trial, with different combinations of experts activated over tasks and time periods. The following sections analyze these dynamics in more detail, especially in comparison to the dynamics observed in the brain. Importantly, for the baseline model, clusters do not seem to employ very different combinations of experts across tasks. The more differentiated usage of pathways across clusters can be seen in the distribution of the numbers of tasks contained in a given cluster (Figure 4, left). Here, our model shows a very distinct power-law distribution of several large clusters containing $> 20$ tasks and many small clusters capturing task-specific pathway

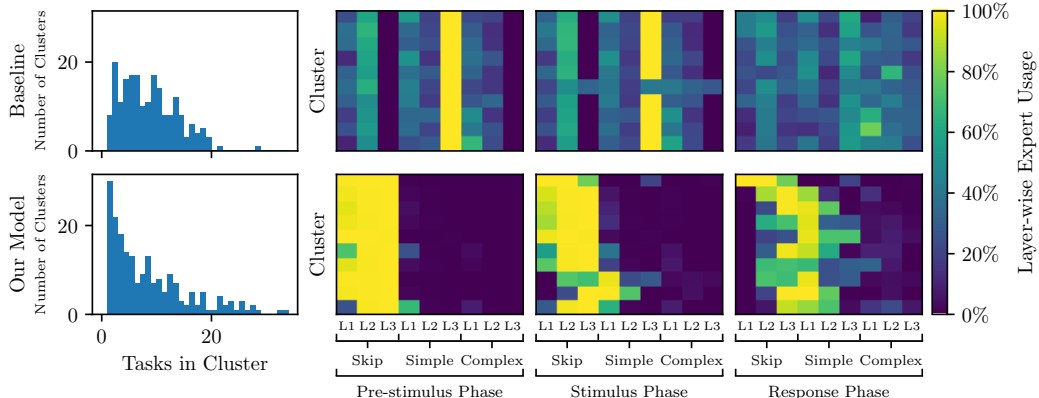

Figure 4: **Task clustering derived from expert usage patterns.** Clusters are averaged over three phases of each task: the pre-stimulus phase, during which the task is known but no input data is presented, the stimulus phase, during which input data is presented, and the response phase. Left: Sizes of clusters across training runs. Right: Average routing weights across training runs by cluster (y-axis) and task phase. In each phase, expert usages within each layer (across each layer's skip connection, simple expert, and complex expert) sum to 100%. The complex expert is rarely used in the first two phases, during which the model outputs zero, but has an average usage as high as 11% in the most complex cluster (see Appendix A.8). L1 / L2 / L3 = Layer 1 / Layer 2 / Layer 3.

usages. The baseline model, on the other hand, seems to distribute the tasks more evenly across clusters, indicating that there are much less distinct routing patterns. This can be shown quantitatively in the sizes of the largest clusters for each model run, which are significantly larger for our model than for the baseline model across 10 different random seeds ($p < 0.0001$). Further visualizations are provided in Appendices A.7 and A.8.

Our results so far show that pathways do not automatically develop from a heterogeneous mixture of experts. Training models with a routing-complexity loss, scaling it based on task performance, and adding expert dropout, all encourage stable pathways to form. These three features define our *Mixture-of-Pathways* (MoP) model. In the following section, we investigate whether the pathways that form in our MoP model mirror the pathways observed within the primate brain.

## 5 Do Artificial Pathways Behave Like the Brain's Pathways?

In the previous section, we showed how our brain-inspired architectural contributions resulted in the formation of a mixture of processing pathways in our model. Now, we will evaluate the degree to which these artificial pathways resemble the behavior of established processing pathways in the brain. Our analyses primarily focus on pathways and dynamics of the brain relating to task difficulty.

### 5.1 Solving Tasks of Varying Difficulty

When analyzing brain activations across tasks with varying levels of difficulty, there is a distinct group of activations in a large frontoparietal network when solving complicated tasks. Since this network is activate while solving any difficult task, it was named the multiple-demand (MD) system [57, 58]. It can be identified both in humans and non-human primates [4, 59]. With this in mind, we now want to test whether the selection of experts used to solve a task is indicative of task difficulty.

To relate these findings from the MD system to our model, we expect that when solving tasks of increasing difficulty, our model should learn to activate increasingly complex regions (schematic in Figure 5). We test this by measuring the correlation between a task's difficulty and the learned pathway complexity ($LPC$) used by the model to solve the task. We quantify the difficulty of a task by the number of training steps it takes a standalone GRU to learn the task (see Appendix A.9 for details and alternative ways of quantifying task difficulty). We find that our full MoP model shows a positive significant relationship, whereas the baseline model does not, matching our expectations.

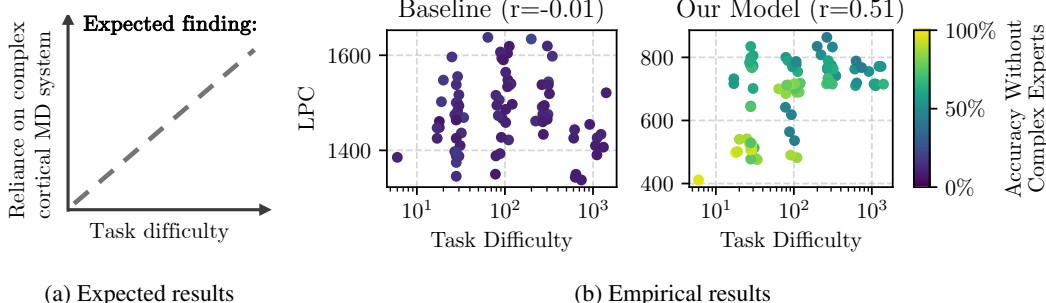

(a) Expected results        (b) Empirical results

Figure 5: **Our model allocates less computation toward solving simpler tasks.** Additionally, when the most complex experts in each layer are disabled, our model is still able to solve the simplest tasks with high accuracy.

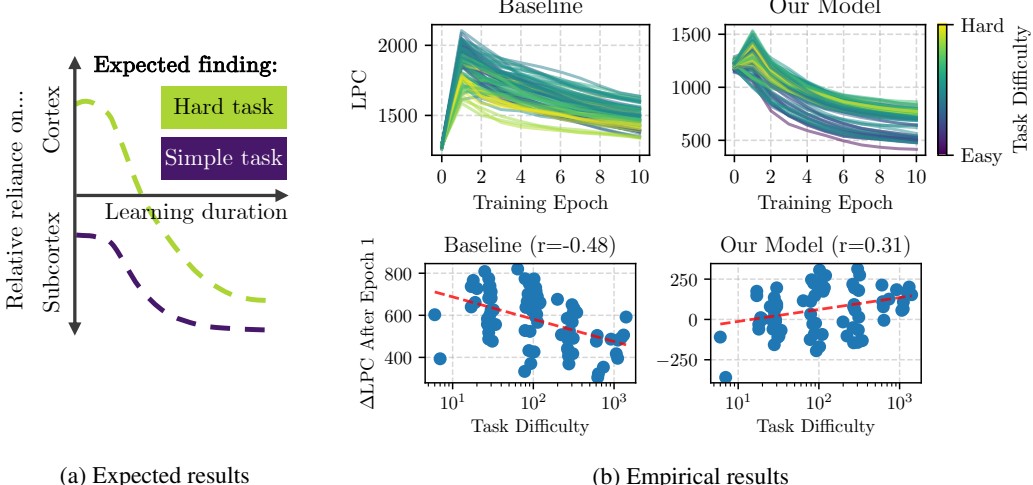

(a) Expected results        (b) Empirical results

Figure 6: **Our model moves complex tasks to more complex pathways at the start of training to support the learning process.** (a) Conceptual schematic of cortical-subcortical interactions. (b) Top row shows the pathway complexity over learning per task averaged across 10 training runs. Lower row shows the change in pathway complexity between model initialization and the end of the first epoch as a function of task difficulty. Our model specifically relies on complex pathways to learn difficult tasks, similar to how the brain relies on complex pathways to support the acquisition of complex skills, even if they are gradually moved to simpler pathways later on.

Furthermore, it is known that patients with lesions to their MD system struggle solving difficult tasks but their ability to solve simple tasks usually is unaffected [60, 61]. We find that the same is true in our models: if the most complex expert in each layer is lesioned, our model can still solve simple tasks with high accuracy, but accuracy on difficulty tasks drops significantly. The same is not true of the baseline model: performance on all tasks drops to near-chance.

## 5.2 Learning Tasks of Varying Difficulty

A more nuanced view of pathways in the brain comes from observing how tasks of varying difficulty are learned over time. Here, an interesting distinction between complex and simple tasks is observed: while simple tasks can be learned through simpler (subcortical) regions alone, complex tasks require more complex (cortical) regions for learning [62–65] (see schematic in Figure 6a). However, as learning continues, even complex task skills are often "transferred" from complex to simple brain regions. This is possible despite the simple pathway not being sufficient to drive the learning process in the first place. We now want to test whether this phenomenon can be observed in our model.

Table 1: Task accuracy and pathway metrics for modified versions of our model.

| Model | Accuracy | Fig. 5 Correlation | Fig. 6b Correlation |
|---|---|---|---|
| Baseline | $91.1\% \pm 8.9\%$ | -0.01 | -0.49*** |
| Our model$^\dagger$ | $83.0\% \pm 15.5\%$ | 0.54*** | 0.31** |
| Without dropout | $90.1\% \pm 8.9\%$ | 0.55*** | 0.03 |
| $\alpha = 1e^{-4}$ | $69.0\% \pm 20.1\%$ | -0.57*** | -0.37*** |
| $\alpha = 1e^{-6}$ | $89.7\% \pm 9.0\%$ | **0.62***** | 0.18 |
| Without task embeddings | $83.0\% \pm 14.2\%$ | 0.58*** | **0.58***** |
| Router dim = 32 | $83.2\% \pm 14.5\%$ | 0.46*** | 0.33** |
| Router dim = 128 | $81.9\% \pm 16.9\%$ | 0.35** | 0.27* |

$\dagger$ With dropout, $\alpha = 1e^{-5}$, task embeddings, and router dim = 64
* $p < 0.05$, ** $p < 0.01$, *** $p < 0.001$

To study this effect, we track the complexities of the learned pathways across tasks over the duration of learning. Figure 6 shows these results across models: our MoP model seems to indeed learn complex tasks by first increasing their pathway complexity, but then reducing it gradually during learning. In contrast, very simple tasks do not increase in pathway complexity at all over learning. We now want to quantify this effect. To translate Figure 6a to our models, we can quantify to which degree the pathway complexity of a given task is increasing or decreasing after the first training epoch. This is a measure of how much the model specifically uses a more complex pathway to learn a given task. Based on findings from neuroscience, we would expect complex tasks to be explicitly moved to more complex pathways, relative to the random starting point, whereas this should not be necessary for simple tasks [35, 62, 66]. Figure 6b shows that this phenomenon can be observed in our model. The more difficult a task is, the more its pathway complexity increases at the start of learning, with the simplest tasks immediately getting routed toward simpler pathways ($r = 0.31$, $p = 0.0040$). This is not true in the baseline model, where we observe the opposite effect: the pathway complexity used to solve the most difficult tasks increases the least at the start of learning ($r = -0.48$, $p < 0.0001$). Upon further inspection, this happens because in an effort to minimize the routing loss, the baseline model learns to push tasks that it is unable to solve toward simpler pathways prematurely. As a result, the baseline model fails to learn the most difficult tasks until the very end of the training process.

## 5.3 Ablations

Lastly, we investigate how changes made to our model can alter the degree to which it resembles pathways in the brain, as discussed in Sections 5.1 and 5.2. Table 1 shows the effects of design parameters on the correlations reported in Figures 5 and 6. Notably, we found that when trained with our loss function that scales based on LPC but without dropout, our model exhibits the effect shown in Figure 5, but not the effect shown in Figure 6. This indicates that the finding in Figure 6 is due to an interaction between the LPC scaling in our loss function and dropout, and can not be explained by training with the LPC regularization alone. We speculate that expert dropout forces the model to be explicit about which pathway is used in learning, and is crucial for creating brain-like learning dynamics. This finding highlights how our model's behavior specifically results from the interplay of all three of our proposed inductive biases.

We also find that removing the task embedding layer improves the finding in Figure 6, however its removal drastically slows down training since the active task is represented with 82 dimensions at each timestep rather than 16 (see Section 3.2). Changing the router's hidden size does not meaningfully affect our results. Increasing or decreasing the penalty for using large experts ($\alpha$) naturally has a meaningful effect on the results, where too strong of a penalty inhibits learning the tasks as well as general convergence of the model, and too weak of a penalty does not sufficiently motivate the model to reduce its usage of complex experts. All rows in the table are averaged over 10 runs.

## 6 Discussion

In this work, we adapted the heterogeneous Mixture-of-Experts architecture to investigate how brain-like processing pathways can form between layers of heterogeneous experts. While these experts do

not form pathways on their own, once trained with a routing-cost loss, task-performance scaling, and expert dropout, we find that they create a *Mixture of Pathways*. Our model provides an account of task-specific brain-wide pathways commonly observed in neuroscience.

**These findings are relevant for neuroscience**, as energetic and processing complexity related priors have been key explanatory mechanisms for how the structure and function of the brain arises [13, 32, 49, 50]. We show that incentivizing models to learn to prioritize simple experts drives the development of brain-like processing pathways from a heterogeneous set of expert models. Additionally, we show how stochasticity of signals is important for learning self-sufficient processing networks. Our model represents an exciting new architecture which can be expanded in the future to study additional heterogeneities present in the connectome, such as varying cell-types and neurotransmitters. Region-specific models, such as those for the hippocampus [67], could be integrated within our architecture.

The brain's implementation of the complexity-guided routing mechanism would likely be found in thalamic nuclei, which regulate information flow between cortical regions [68]. Two systems could modulate pathway selection based on metabolic costs: norepinephrine release from the locus coeruleus, which controls cognitive effort and processing power allocation [69–71], and hypocretin/orexin neurons in the hypothalamus, which govern metabolic resource budgets [72]. Both systems project strongly to thalamic nuclei and could influence routing decisions between simple and complex processing pathways. This suggests our model's routing-cost mechanism may reflect how the brain balances computational demands against metabolic constraints through neuromodulatory control of thalamocortical interactions. While mapping our router onto a specific brain region might feel natural, it should be added that mechanisms like predictive coding can implement routing and filtering operation between regions without the need of an explicit router region [73].

**In the context of machine learning**, our work builds on the recent widespread adoption of the Mixture-of-Experts architecture for building parameter-efficient large language models [15, 74]. Recent innovations specifically aim at allowing MoE models to process queries dynamically to reduce processing costs [17]. Our small-scale simulations show how it may be possible to use heterogeneous experts alongside a processing-cost loss function that allows the model to dynamically allocate resources to processing tokens. Finally, load balancing in MoEs prevents over-reliance on a single expert by encouraging distributed processing [15]. Our complexity loss serves as a task-driven form of load balancing.

## 6.1 Limitations and Extensions

There are several ways to expand our investigations. **On the neuroscience side**, our architecture introduces a new way of modeling multi-region interactions of the brain, but some key architectural characteristics are not yet taken into account. Most importantly, the primate brain has large loop structures which would allow signals to return to a region [27]. Our architecture only allows a forward progression of signal and does not allow signals to be routed back to earlier layers. At the same time, while our analyses demonstrated a link between experts and cortical and subcortical regions, we have not linked the router component of our HMoE layers to a specific component of the brain. Potential options are discussed earlier in Section 6, we do not make any explicit comparison to brain data yet. **On the ML side**, our training setup currently focuses on solving relatively simple tasks with a small model. To see whether our complexity-aware routing and load-balancing measures scale to larger networks, we would need to train larger models on more difficult tasks. Lastly, **on the identification of pathways**, we currently rely on three independent tests to see whether a model contains pathways. Future investigations would ideally identify one specific metric to quantify the degree to which a Mixture-of-Experts architecture has formed pathways.

## 7 Conclusion

In this paper, we introduced a modified Heterogeneous Mixture-of-Experts architecture that results in the formation of recognizable processing pathways. Analysis of these pathways during learning and problem solving revealed a similarity between these pathways and those observed in the brain. Our new *Mixture-of-Pathways* architecture serves as a new theoretical tool for neuroscience and can guide the search for future resource-efficient architectures in machine learning.

## Acknowledgments and Disclosure of Funding

We thank our reviewers, the *Neural & Machine Learning Group* at the University of Oxford, and the *AI HW SW CoDesign Workstream* at the Open Compute Project (OCP) for helpful feedback. This research is supported by the EPSRC (EP/X029336/1) and an ERC-UKRA Frontier Research Guarantee Starting Grant (EP/Y027841/1) awarded to R.P.C. J.C.'s work was supported by a Rhodes Scholarship. J.A.'s work was partially supported by a Career Development Research Fellowship from St John's College, Oxford. We additionally thank Modal for compute credits.

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

# A Appendix

## A.1 Implementation Details and Code Access

While most details on the model implementation are described in Section 3, we provide some additional details here. The GRUs in all of our layers, including routers, use the ReLU activation function and are initialized from $\mathcal{U}(-\sqrt{k}, \sqrt{k})$, where $k$ is the GRU's hidden size, as is standard in PyTorch. All models are optimized with the Schedule-Free variant of the AdamW optimizer [47] using a learning rate of 0.01, betas of $(0.9, 0.999)$, and no weight decay. As briefly described in Section 3, we use an additional embedding layer to transform the 82-dimensional one-hot encoding of the task identity into a 16-dimensional embedding vector, which is concatenated with the task stimuli before being provided to the model. We include these details in a complete version of our training algorithm below in Algorithm 2, expanding on the abbreviated algorithm introduced in Algorithm 1. We provide our implementation at https://github.com/jackcook/mixture-of-pathways.

---

**Algorithm 2:** Full Mixture-of-Pathways training protocol.

---

**Data:** Task dataset $D$, Experts $E = \{e_1, e_2, \ldots, e_n\}$, Routers $R = \{r_1, r_2, \ldots, r_n\}$

Initialize routers and heterogeneous experts, sampling weights and biases from $\mathcal{U}(-\sqrt{k}, \sqrt{k})$;

Process 82-dimensional one-hot task encoding with embedding layer;

Set $h_0$ to task input, consisting of a 1-dimensional fixation input, two 16-dimensional stimuli, and a 16-dimensional task embedding at each timestep;

**for** *each training step* **do**

    Sample batch $b$ from $D$;

    **for** *each layer $l$* **do**

        Compute routing weights $w_{l,j} = \text{softmax}(r_l(h_{l-1}))$ ;

        Apply expert dropout (Not in baseline architecture; see Section 4.2) ;

        Compute expert activations $w_{l,j}$ for each expert in $l$ ;

        Combine outputs: $h_l = \sum_j w_{l,j} \cdot h_{l,j}$;

    **end**

    Compute baseline model losses: $L_{\text{fix}}$ and $L_{\text{response},i}$ ;

    Compute $L_{\text{routing}}$ loss (Not in baseline architecture; see Section 4.1) ;

    Compute $L_{\text{total}} = L_{\text{fix}} + L_{\text{routing}} + \sum_i L_{\text{response},i}$;

    Update parameters using Schedule-Free AdamW optimizer [47];

**end**

---

## A.2 Sample Task Visualizations and Descriptions

Section 3 briefly described the Mod-Cog task set [24]. Here, we provide a more detailed description of the tasks, alongside visualizations of sample trials.

The Mod-Cog task set consists of 82 time-series-based cognitive tasks that extend the original NeuroGym framework [48] through two primary modifications: integration tasks, which incorporate interval estimation based on delay periods, and sequence generation tasks, which require time-varying outputs with drifting directions. The original 20 cognitive tasks from NeuroGym serve as the foundation, with new integration-based tasks and new sequence generation tasks forming a set of 82 tasks in total. These tasks span a wide range of cognitive demands, from simple stimulus-response mappings to complex working memory and sequential decision-making challenges.

Tasks are presented as continuous time-series data, which we sample at 100-millisecond intervals. Each input consists of a 1-dimensional fixation signal, two 16-dimensional stimulus channels, and an 82-dimensional one-hot task identifier that passes through a learned embedding layer, as described in Appendix A.1. During the fixation period of each task, models must maintain an output of zero while processing incoming stimuli. During the subsequent response period, models must return task-specific output sequences. Task difficulty varies systematically according to several factors: the complexity of decision rules (from simple detection to multi-step integration), the duration of delay periods that tax working memory, the number of stimuli that must be simultaneously tracked, and whether responses require static outputs or dynamic sequential patterns. Figure 7 illustrates six representative tasks that demonstrate this range of complexity.

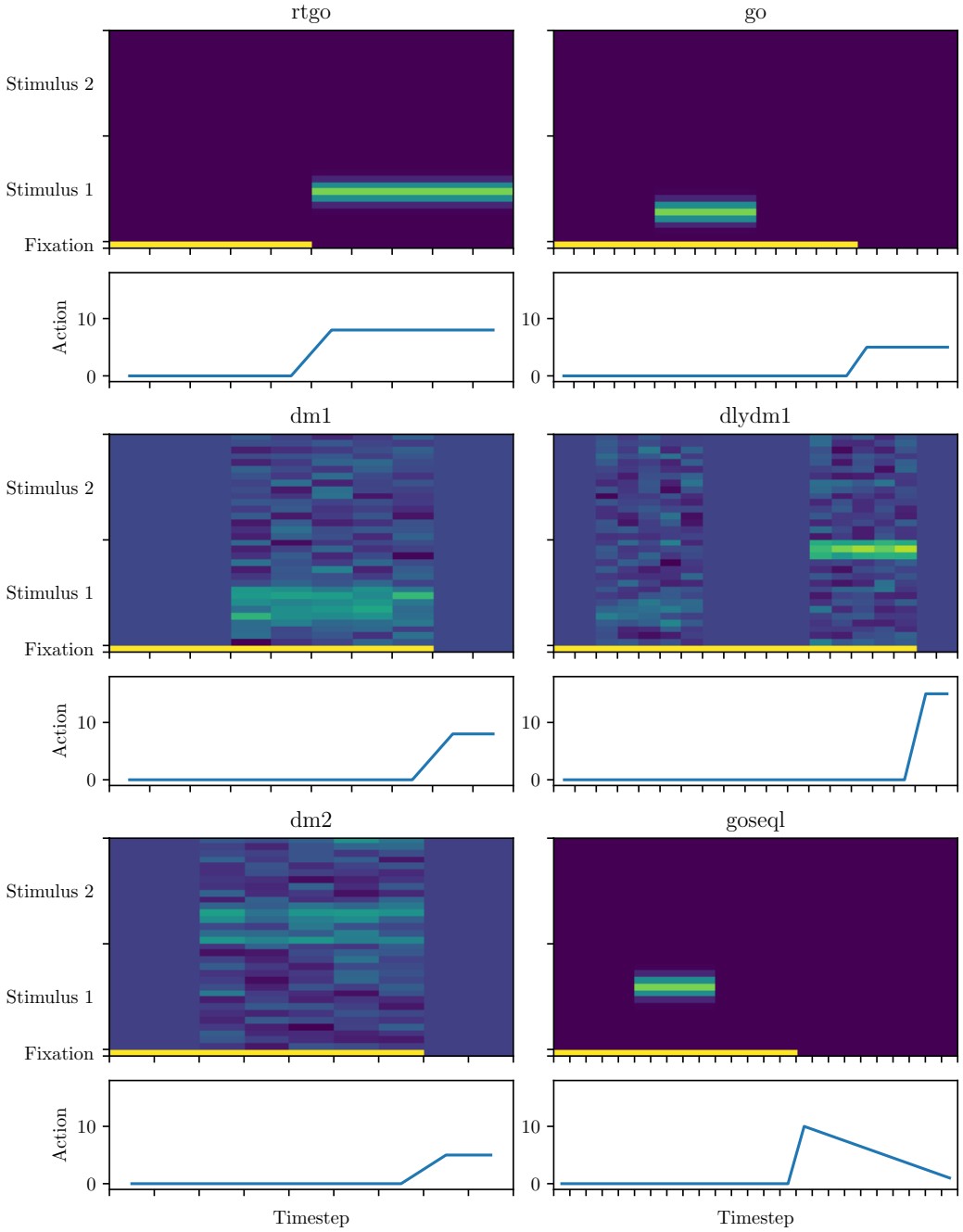

Figure 7: Model inputs and actions for the `rtgo`, `go`, `dm1`, `dlydm1`, `dm2`, and `goseqr` tasks. Appendix A.2 provides a detailed description of each task.

- **RTGo task** (`rtgo`): The model must immediately output the value presented in either input channel.
- **Go task** (`go`): The model observes two input channels and must respond with the value presented in either stimulus once the fixation period has ended.
- **Decision Making task** (`dm1`): The model observes brief stimulus presentations from both channels and must respond with the value in stimulus 1 that had the highest average intensity (values in stimulus 2 can be ignored). This requires integration and comparison of sensory evidence.
- **Delayed Decision Making task** (`dlydm1`): Similar to dm1, but includes a delay period between a decoy stimulus and the noisy stimulus, requiring working memory to maintain stimulus information.
- **Dual Decision Making task** (`dm2`): Similar to dm1, but the model must return the highest value from stimulus 2.
- **Sequential Decision Making task** (`goseql`): Based on the Go task but requiring a time-varying output that drifts in a specific direction (leftward) over the response period, combining stimulus detection with sequential motor control.

## A.3 Loss Variants Used Across Model Types

In Section 4, we introduce architectural changes to the model that encourage pathways to form between layers. Here, we give an overview of the five different loss functions which are used to train our different model variants.

The loss function that we use to train the baseline model only includes the fixation and task response loss, as shown below in Equation 4. Equation 5 computes a mean-squared error for the fixation period where the correct output value is always zero. This loss function is not task-specific since it can be computed across all task inputs while the fixation input is active. Equation 6 computes a task-specific cross-entropy loss between the 16 possible output values and the model's 16 output logits at each timestep. When used to train a model, one of these response losses will be computed for each task in the set of tasks $\mathcal{T}$.

$$L_{\text{Baseline}} = L_{\text{fix}} + \sum_i^{\mathcal{T}} L_{\text{response},i} \tag{4}$$

$$L_{\text{fix}} = \frac{1}{T} \sum_t^{T} \hat{y}_t^2 \tag{5}$$

$$L_{\text{response},i} = -\frac{1}{T_i} \sum_t^{T_i} y_t \log(\hat{y}_t) \tag{6}$$

Here, $\hat{y}_t$ represents the model's output logits at timestep $t$ during fixation, $T$ is the total number of fixation timesteps, $y_t$ is the true target output, $\hat{y}_t$ is the model's predicted output at timestep $t$, and $T_i$ represents the total number of response timesteps for task $i$.

In Section 4.1, using this baseline loss function, we find that the baseline model by itself does not converge on consistent pathways across model runs. This motivates us to create a new loss function, $L_{\text{RC}}$, which introduces a routing cost that penalizes the model for using more complex experts, as shown in Equation 7.

$$L_{\text{RC}} = L_{\text{fix}} + \sum_i^{\mathcal{T}} (L_{\text{response},i} + \alpha LPC_i) \tag{7}$$

$$LPC_i = \frac{1}{T_i} \sum_t^{T_i} \sum_j^{E} w_{i,j,t} s_j^2 \tag{8}$$

In Equation 8, $w_{i,j,t}$ represents the routing weight assigned to expert $j$ at timestep $t$ for task $i$, $s_j$ is the size of expert $j$, and $E$ denotes the total number of experts in the model (in this manuscript, all models have three layers of three experts each, so $E = 9$). $\alpha$ is a hyperparameter that balances the trade-off between the model's performance on each task and the complexity of the experts used to solve that task. If $\alpha$ is too large, the model will reach a local minimum where it is unable to solve any task, but it can reduce its expert usage to zero by using the skip connections in each layer and performing no computation. On the other hand, if $\alpha$ is too small, the model will solve each task to a very high degree of accuracy, but not reduce the complexity of the experts used to solve each task, and fail to form brain-like pathways. We found that setting $\alpha = 10^{-5}$ balanced these priorities well, and used this value for all of the experiments in this work. However, future work may investigate a better method for selecting this hyperparameter.

While the routing consistency for the model with the loss in Equation 7 is improved as shown in Figure 2, we do find that it does not converge on a fully consistent routing pattern. This motivates the addition of a scaling factor based on task performance, which reduces the effect of the routing loss when task performance is low. The resulting loss shown in Equation 9 is the final loss we use to train our *Mixture-of-Pathways* model.

$$L = L_{\text{fix}} + \sum_i^{\mathcal{T}} (L_{\text{response},i} + \frac{\alpha LPC_i}{L_{\text{response},i} + \epsilon}) \tag{9}$$

The normalization term $L_{\text{response},i} + \epsilon$ uses the task-specific response loss $L_{\text{response},i}$ to scale the routing penalty, where $\epsilon$ is a small constant added to prevent division by zero when the model achieves perfect task performance.

### A.4 Learned Pathway Complexity and Routing Consistency

#### A.4.1 Calculation Example

Here we provide a detailed example calculation of how the expert size penalty is calculated. This penalty is used within the calculation of the routing consistency described in Section 4.1 and also as part of the expert-usage loss from Equation 7.

In this equation, $E$ is the set of all experts, $s_j$ is the size of each expert, and $w_{i,j}$ is the weight assigned to expert $j$ while solving task $i$. For example, imagine a model with three experts: a skip connection, a simple expert with 16 neurons, and a complex expert with 32 neurons. To solve a task $i$, imagine the model sets the weight of the skip connection to $31\%$, the simple expert to $43\%$, and the complex expert to $26\%$. The model's learned pathway complexity (LPC) for task $i$ would be 376.3, as follows:

$$LPC_i = w_{i,0}s_0^2 + w_{i,1}s_1^2 + w_{i,2}s_2^2 = (0.31)(0)^2 + (0.43)(16)^2 + (0.26)(32)^2 = \boxed{376.3} \tag{10}$$

For simplicity, this equation shows how to calculate the model's LPC at a single timestep. To calculate the LPC for an entire task, this value should be calculated at and averaged over all of the task's timesteps.

#### A.4.2 Routing Consistency Across Model Types

Using the calculated LPC values for each task, we determine the consistency of the routing decisions made by different models when solving tasks. In Figure 8, we show a version of Figure 2 with our final model, which includes expert dropout. After the addition of expert dropout, the mean pairwise correlations of the models' routing consistency is 0.51 ($p < 0.0001$). This is a reduction when compared to our model trained only with our custom routing cost and task performance scaling, however, as explained in Section 4.2, the model with the additional dropout does develop self-sufficient pathways on top of the consistency criteria, so that the model including the dropout overall better matches the pathway formation criteria. Our main investigation on what we call the *Mixture-of-Pathways* model includes the expert dropout.

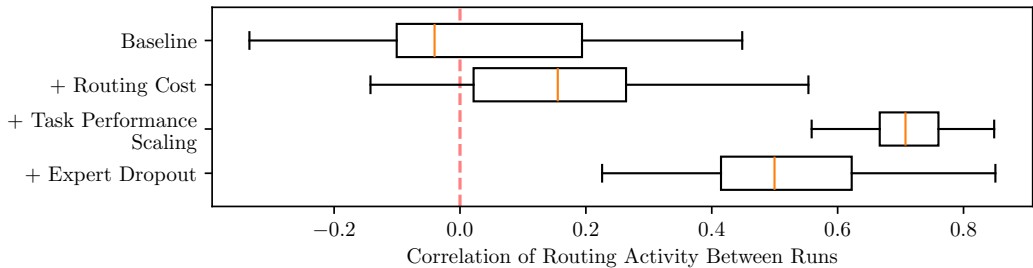

Figure 8: Routing consistency as measured by correlation of routing activity across model runs. This is a version of Figure 2 which also shows the model with expert dropout. Even with expert dropout the routing consistency is significantly above the baseline model ($p < 0.0001$), even though it is slightly reduced when compared to the model with routing cost and task performance scaling.

## A.5  Effective Rank

To define the $LPC$, in Equation 1 we use $s_j^2$ as a penalty for using expert $j$, where $s_j$ is the expert's hidden dimension, or zero in the case of a skip connection. Intuitively, this was motivated by the $O(s_j^2)$ cost of storing each expert's weight matrix in memory, however this only roughly captures the learning capabilities of a GRU with $s_j$ neurons. For example, as demonstrated by the lottery ticket hypothesis, it is possible that experts with large hidden dimensions may converge on low-rank solutions more easily than experts with small hidden dimensions [75].

To ensure that we were appropriately penalizing experts relative to each other, we analyzed the effective rank of each expert's matrix, defined as the participation ratio of the squared sum of singular values to the sum of squared singular values, $\frac{(\sum \sigma_i)^2}{\sum \sigma_i^2}$, over the course of training [76]. This metric measures how evenly distributed the singular values are and thus how many dimensions the matrix effectively uses. These are shown in Figure 9. For both models, we find that the effective rank of large experts is roughly double that of small experts, both before and after training, so that large experts always have a much larger effective rank than small experts ($p < 0.0001$). We additionally find that all experts across both our model and the baseline model decrease slightly in effective rank over training, but the differences between these models' effective ranks tends to be small and never becomes significant ($p > 0.05$). This supports the conclusion that the hidden dimension is at least a good approximation of processing complexity, but we encourage future work to consider measuring this with a scalar value.

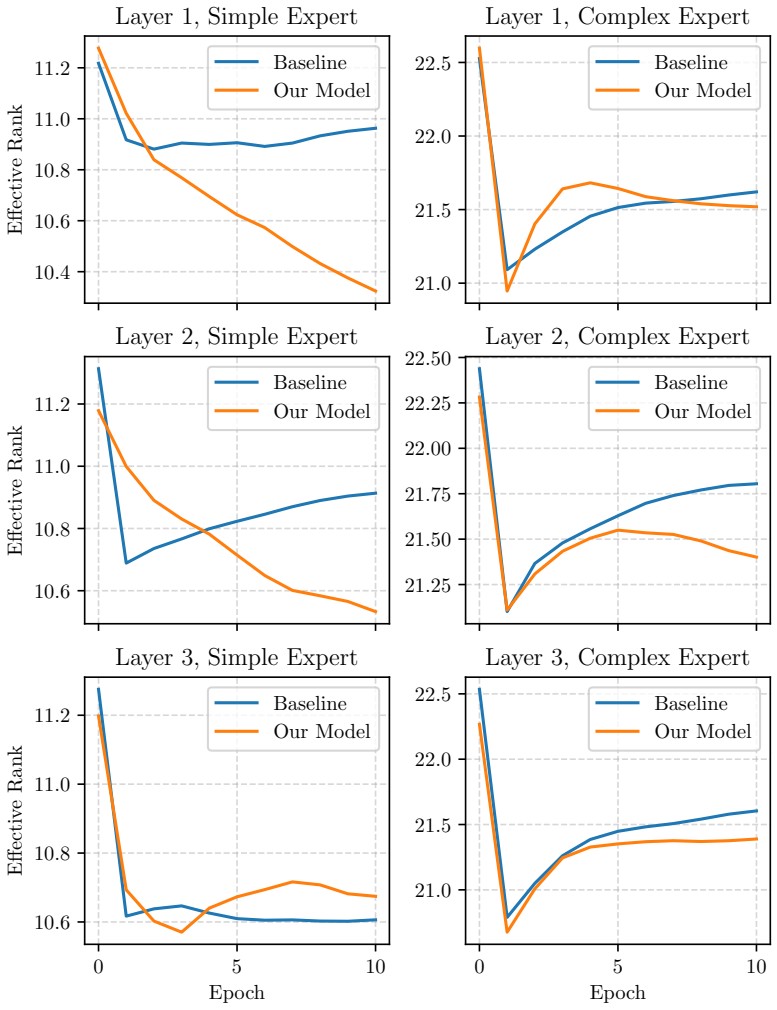

Figure 9: Changes in the effective rank of each expert's weight matrix over the course of training. Results for each configuration are averaged across 20 runs.

## A.6  Per-Task Accuracy Metrics

In Table 2 below, we report accuracy metrics on each task for three models: the baseline HMoE model described in Section 3.1, our model, which includes the routing cost and task-performance scaling described in Section 4.1 and the expert dropout described in Section 4.2, and our model without expert dropout.

Table 2: Per-task accuracy metrics.

| Task | Baseline | Our Model | Without Dropout |
|---|---|---|---|
| **Mean** | 91.1% | 83.0% | 90.1% |
| **Median** | 93.3% | 87.5% | 91.1% |
| anti | 100.0% | 100.0% | 99.9% |
| antiseql | 100.0% | 99.8% | 99.8% |
| antiseqr | 100.0% | 100.0% | 100.0% |
| ctxdlydm1 | 99.5% | 99.6% | 99.5% |
| ctxdlydm1intl | 99.8% | 100.0% | 99.3% |
| ctxdlydm1intr | 99.6% | 98.4% | 99.6% |

| Task (cont.) | Baseline | Our Model | Without Dropout |
|---|---|---|---|
| ctxdlydm1seql | 83.5% | 79.9% | 84.0% |
| ctxdlydm1seqr | 83.2% | 84.1% | 82.3% |
| ctxdlydm2 | 75.5% | 71.7% | 76.3% |
| ctxdlydm2intl | 78.2% | 70.9% | 79.3% |
| ctxdlydm2intr | 81.2% | 82.3% | 84.2% |
| ctxdlydm2seql | 95.4% | 90.2% | 94.3% |
| ctxdlydm2seqr | 90.7% | 92.7% | 90.6% |
| ctxdm1 | 93.0% | 90.0% | 93.7% |
| ctxdm1seql | 93.6% | 88.6% | 91.0% |
| ctxdm1seqr | 82.2% | 83.0% | 76.4% |
| ctxdm2 | 98.7% | 97.9% | 99.3% |
| ctxdm2seql | 99.5% | 97.5% | 98.5% |
| ctxdm2seqr | 99.4% | 92.1% | 99.0% |
| dlyanti | 99.7% | 98.4% | 99.5% |
| dlyantiintl | 99.8% | 39.1% | 98.7% |
| dlyantiintr | 99.6% | 44.3% | 94.1% |
| dlyantiseql | 96.2% | 38.4% | 98.4% |
| dlyantiseqr | 98.7% | 37.6% | 93.4% |
| dlydm1 | 91.8% | 87.5% | 88.6% |
| dlydm1intl | 93.0% | 88.5% | 91.8% |
| dlydm1intr | 91.7% | 89.2% | 88.7% |
| dlydm1seql | 92.8% | 86.0% | 89.3% |
| dlydm1seqr | 90.5% | 90.6% | 88.8% |
| dlydm2 | 90.2% | 87.0% | 90.5% |
| dlydm2intl | 92.2% | 84.1% | 86.8% |
| dlydm2intr | 92.1% | 85.1% | 88.1% |
| dlydm2seql | 82.4% | 78.0% | 74.3% |
| dlydm2seqr | 82.1% | 78.9% | 76.8% |
| dlygo | 80.2% | 59.9% | 76.0% |
| dlygointl | 78.5% | 60.8% | 79.6% |
| dlygointr | 82.6% | 61.5% | 84.3% |
| dlygoseql | 85.5% | 60.4% | 82.0% |
| dlygoseqr | 74.3% | 62.6% | 76.8% |
| dm1 | 76.3% | 61.7% | 75.0% |
| dm1seql | 78.8% | 57.3% | 78.4% |
| dm1seqr | 81.0% | 63.4% | 76.6% |
| dm2 | 100.0% | 98.0% | 99.9% |
| dm2seql | 99.7% | 97.6% | 99.2% |
| dm2seqr | 100.0% | 98.0% | 99.9% |
| dmc | 99.9% | 95.5% | 99.8% |
| dmcintl | 99.8% | 99.0% | 99.4% |
| dmcintr | 99.5% | 98.0% | 99.8% |
| dmcseql | 81.9% | 75.2% | 81.1% |
| dmcseqr | 81.5% | 75.0% | 81.7% |
| dms | 75.2% | 66.5% | 77.2% |
| dmsintl | 73.4% | 64.8% | 77.3% |
| dmsintr | 79.7% | 72.8% | 86.0% |
| dmsseql | 95.1% | 87.6% | 92.8% |
| dmsseqr | 95.2% | 88.6% | 90.6% |
| dnmc | 93.2% | 87.3% | 88.7% |
| dnmcintl | 93.1% | 86.2% | 89.7% |
| dnmcintr | 80.8% | 79.0% | 77.1% |
| dnmcseql | 98.5% | 93.0% | 98.9% |
| dnmcseqr | 99.7% | 94.3% | 98.3% |
| dnms | 98.8% | 91.3% | 98.8% |
| dnmsintl | 99.3% | 93.6% | 98.1% |
| dnmsintr | 99.9% | 99.4% | 100.0% |

| Task (cont.) | Baseline | Our Model | Without Dropout |
|---|---|---|---|
| dnmsseql | 100.0% | 97.4% | 99.6% |
| dnmsseqr | 100.0% | 99.1% | 99.4% |
| go | 100.0% | 94.9% | 99.7% |
| goseql | 99.9% | 96.7% | 99.4% |
| goseqr | 100.0% | 94.0% | 99.7% |
| multidlydm | 81.8% | 74.5% | 81.9% |
| multidlydmintl | 78.1% | 77.3% | 80.0% |
| multidlydmintr | 74.1% | 66.7% | 76.0% |
| multidlydmseql | 73.1% | 66.6% | 71.8% |
| multidlydmseqr | 79.7% | 71.6% | 79.4% |
| multidm | 94.0% | 90.0% | 90.7% |
| multidmseql | 94.3% | 88.9% | 93.1% |
| multidmseqr | 93.3% | 85.9% | 92.6% |
| rtanti | 92.6% | 86.1% | 91.2% |
| rtantiseql | 82.2% | 79.5% | 81.6% |
| rtantiseqr | 99.7% | 95.2% | 98.9% |
| rtgo | 98.7% | 94.9% | 98.9% |
| rtgoseql | 99.3% | 91.9% | 98.0% |
| rtgoseqr | 98.1% | 93.1% | 99.0% |

## A.7 Task-Specific Expert Usage Patterns

In Figures 10 and 11, we show two samples of tasks completed by our model. At each timestep, we plot the model's decisions in orange, which overlap with the blue ground truth values, indicating that in these trials, the model always returned the correct answer. We additionally plot the expert usage of the model at each timestep. At each timestep, three bars are shown for each layer, with their color indicating the usage of the skip connection, the simple expert, and the complex expert, in that order. In the go trial shown in Figure 10, the model primarily uses the skip connections at each timestep during the fixation period, in which it only needs to return zero. During the response period, the model moves its computations toward a more complex pathway, primarily using the simple experts in layers 1 and 2, and the skip connection in layer 3. In the more complicated dmcintr trial shown in Figure 11, the model switches between pathways multiple times depending on its needs, which vary between working memory and computation. This shows that our model has formed distinct pathway and modes of processing information which it dynamically switches between. As a result, the model opens up the possibility of analyzing the detailed dynamics of how pathways are combined over tasks with time courses that include varying computational demands, to learn which principles underlie such coordination processes.

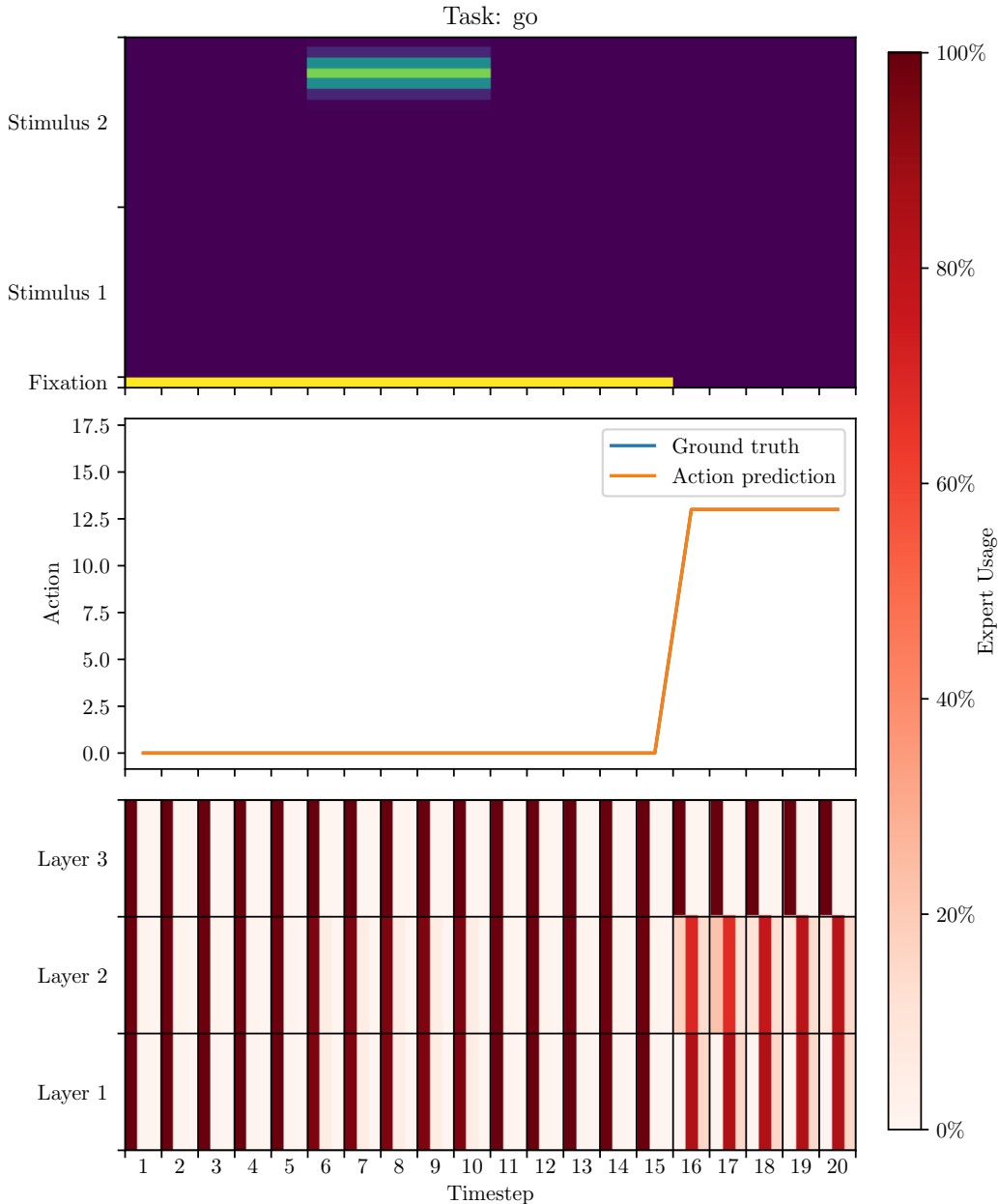

Figure 10: In the go task we see that models rely on extremely simple pathways for their decision making until activating model complex pathways during the response period.

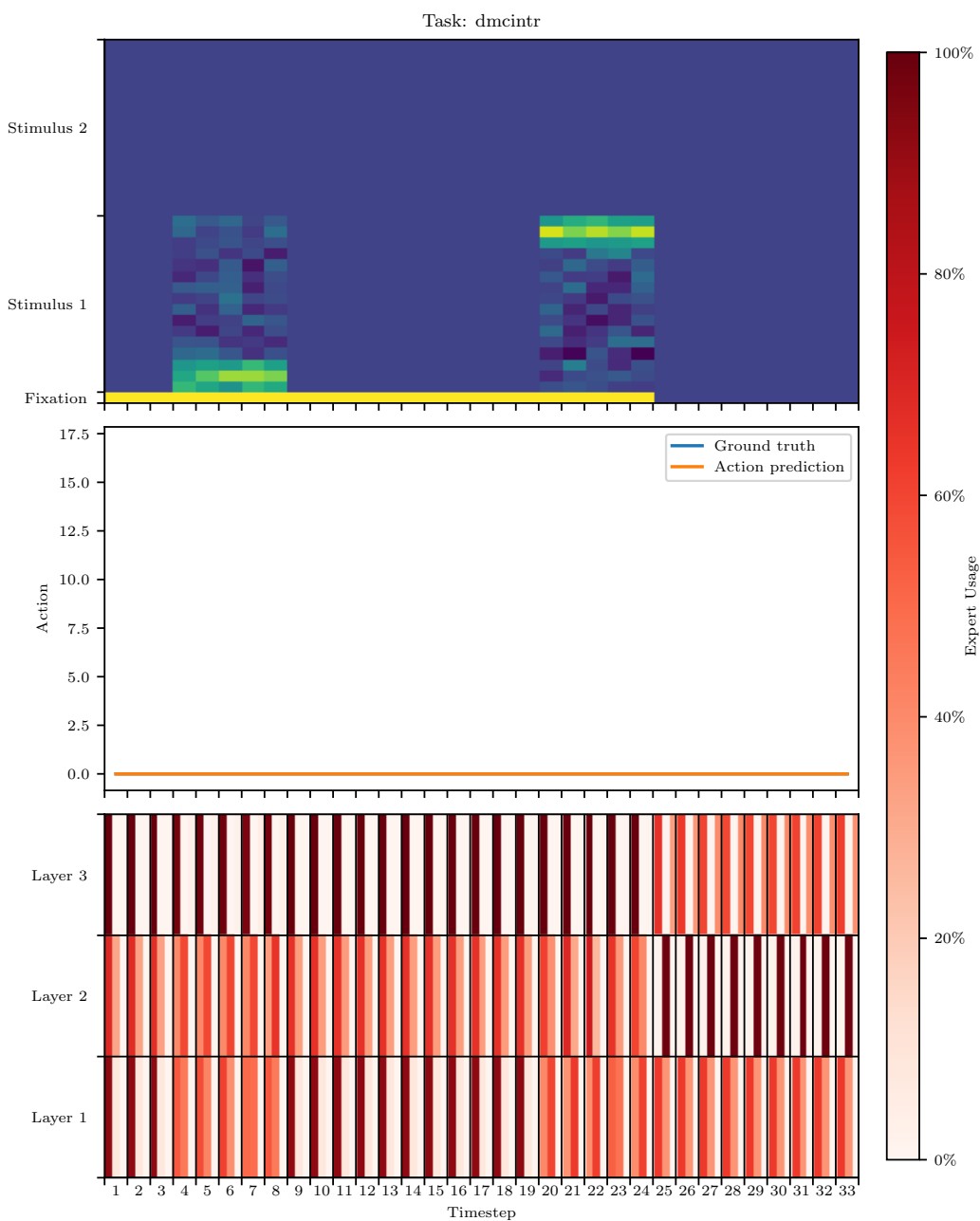

Figure 11: During more complex tasks such as dmcintr we see that models us a a very dynamic and rich set of pathways throughout the duration of a trial to return the correct response.

## A.8 Unclustered Expert Usage Patterns Across Tasks

In Figures 13 and 14, we show the unclustered expert usages during three phases of each task. Each task has a different number of timesteps, so in order to condense expert usage into a single figure, we averaged expert usage over three phases which are shared by each task: a pre-stimulus phase, during which the task identity is known but input data has not yet been presented, a stimulus phase, during which the model is observing input data, and a response phase, during which the model needs to output its responses. In these figures, tasks are sorted based on the same clusters shown in Figure 4 for visual clarity. Notably, the complex expert is rarely used in the first two phases, during which the model outputs zero, but is commonly used during the response phase of complex tasks. When analyzing the average usage of the most complex experts over the clusters shown in Figure 4, we find that the cluster with the highest reliance of the most complex experts uses those with an average routing weight of 0.11. In Figure 12 we show the distribution of complex expert usage by layers of the model, over clusters derived in Figure 4, meaning each data points here is one of the 10 clusters.

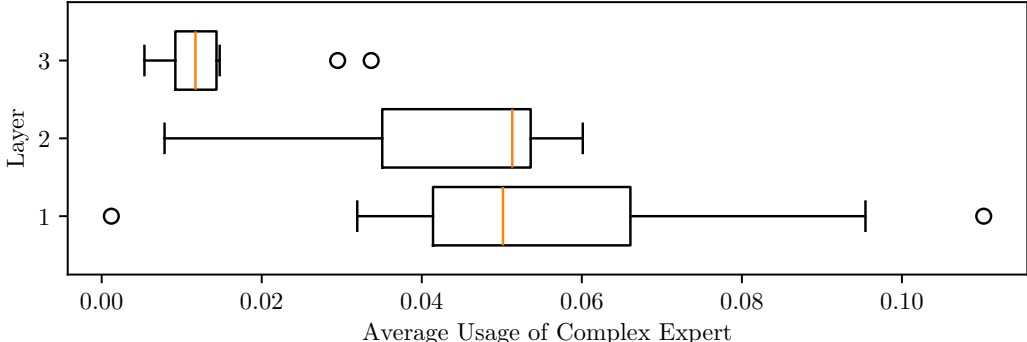

Figure 12: Average usage of the most complex experts for each cluster derived in Figure 12, split by layer index.

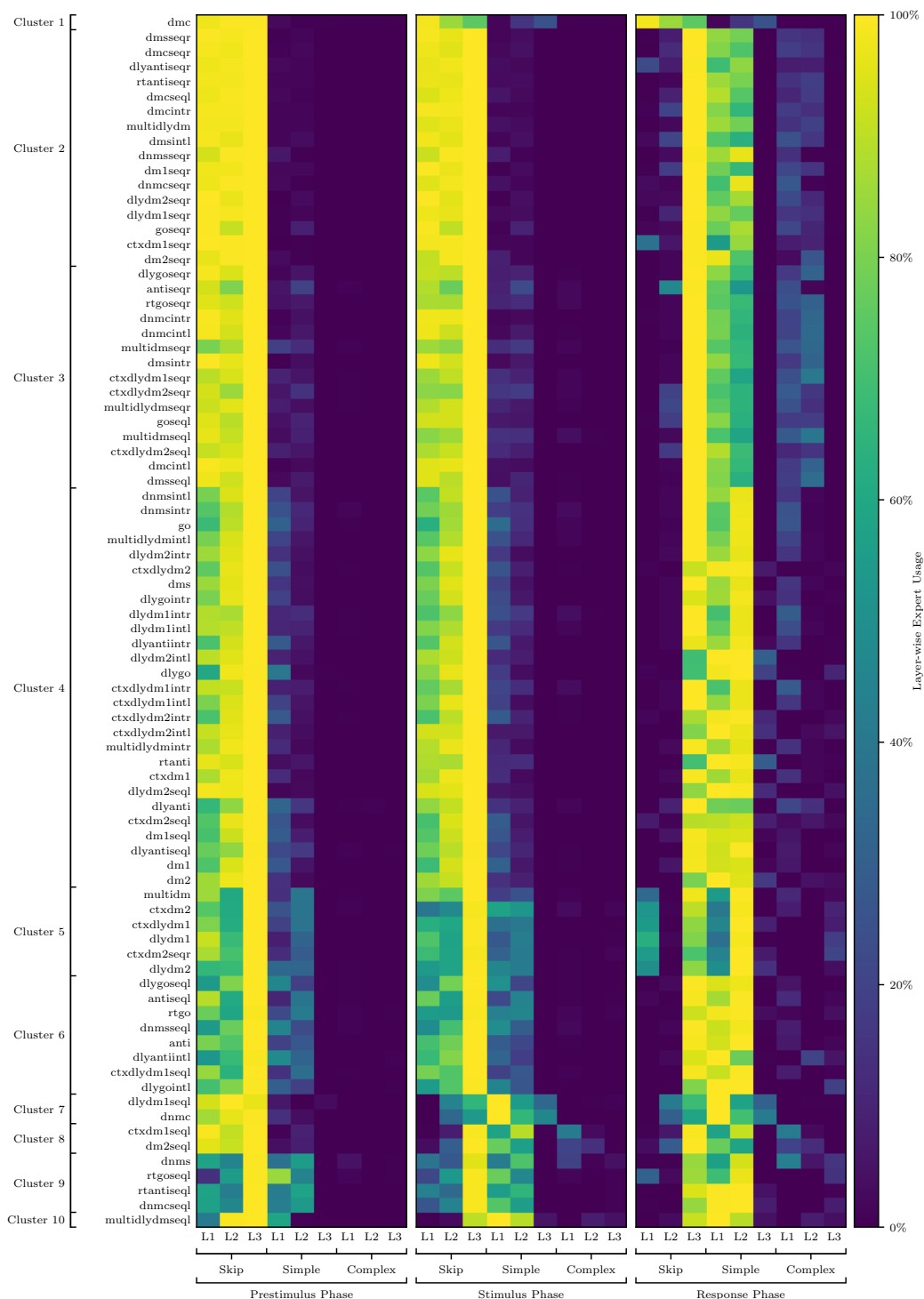

Figure 13: Layer-wise expert usage averaged over three phases of each task: the pre-stimulus phase, during which the task is known but no input data has been given to the model, the stimulus phase, during which input data is being given to the model, and the response phase, during which the model needs to calculate and return the correct response. In each phase, expert usages sum to 100% within each layer, i.e. usages of the skip connection, simple expert, and complex expert of layer 1, denoted by "L1", add up to 100%. Tasks are grouped into 10 clusters, shown along the left, based on similarities in their routing patterns. A contrasting version of this figure for a baseline model trained without our cost-based loss and dropout is shown in Figure 14.

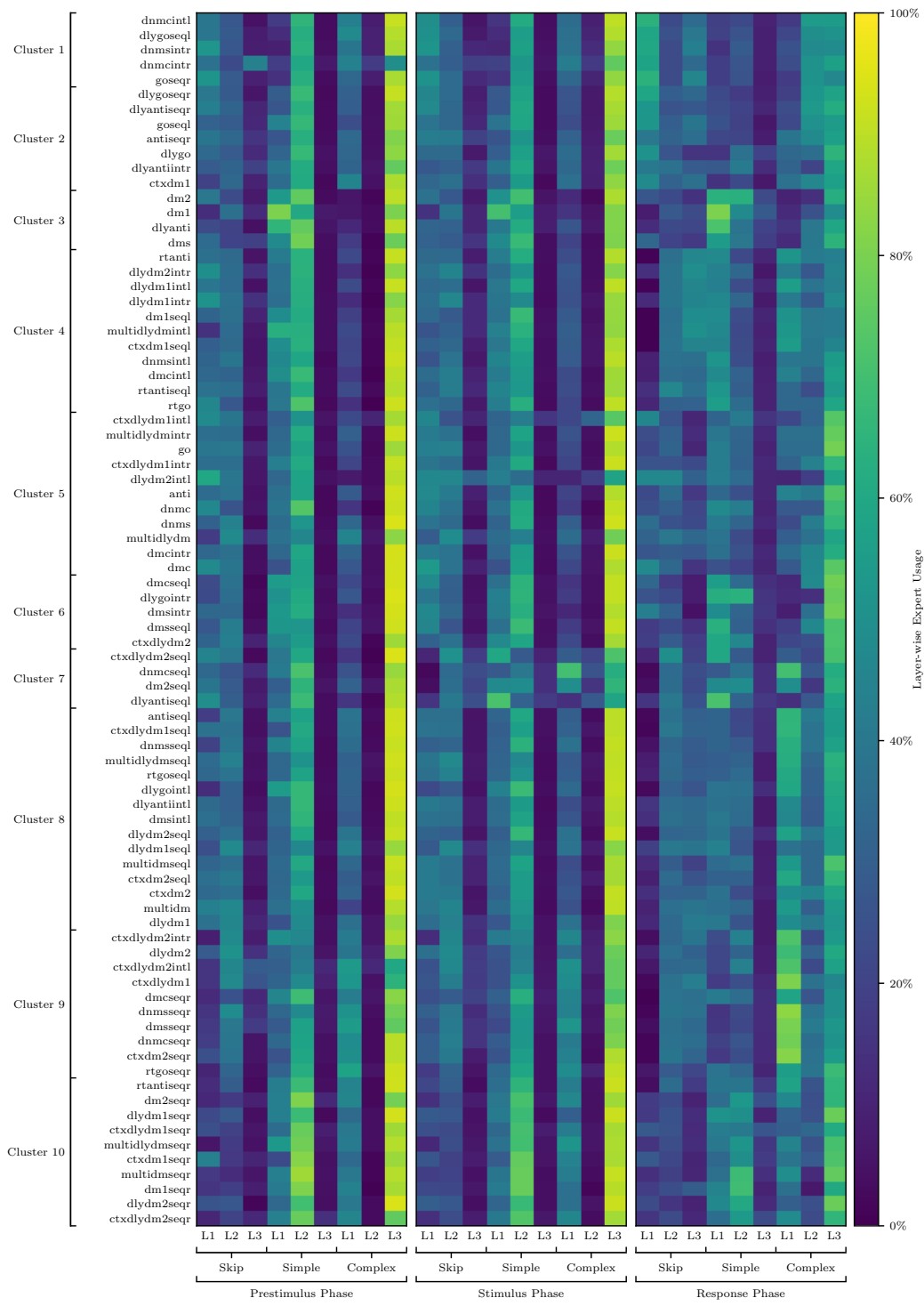

Figure 14: A separate version of Figure 13 for a model trained without our custom routing cost, loss normalization, and expert dropout. The pathways that form are much less distinct, and are also much less stable.

### A.9 Task Difficulty

#### A.9.1 Number of Training Steps

To measure each task's difficulty, we train five recurrent neural networks (GRUs), each with 64 neurons, and record how many training steps it takes each model to solve that task to 99% accuracy. The task's difficulty is then reported as the median number of steps from these five training runs. Figure 15 shows these measurements for all 82 tasks in the Mod-Cog task suite [24].

#### A.9.2 Number of Rules

There are many ways to characterize the difficulty of learning problems [77], but no universal complexity measure has been developed to date. We believe "training steps needed to learn the task," as discussed above in Appendix A.9.1, is a sensible measure because this can be measured without any implicit biases, and takes into account task demands such as working memory and any other factors which make inference challenging [77]. At the same time, it remains unclear whether such a complexity measure would neatly map onto what humans or animals perceive to be "difficult tasks," which is often linked to the number of rules in a task [78]. However, this can be easily tested, as Mod-Cog tasks are created based on combinations of different motifs and rules. For example, Figure 7 shows that the "Delayed Decision Making" task (`dlydm1`) is an altered version of the standard "Decision Making" task (`dm1`) with the added "Delay" rule (`dly`).

We find that our difficulty metric is in fact correlated with the number of rules in each task ($r = 0.39$; $p < 0.0001$, shown in Figure 16), and that our model exhibits an even stronger correlation for the brain-like finding in Figure 5 when this is used as the difficulty metric ($r = 0.57$, $p < 0.0001$) compared to the baseline model ($r = -0.09$, $p = 0.4288$). At the same time, using "number of rules" as the actual difficulty metric has two downsides: (a) it is a discrete and ordinal measurement from 1 to 4 with less statistical power, and (b) some rules are harder to learn than others (i.e. `go` vs. `dm1`). This suggests to us that our current convergence-based complexity measure is, at least in this specific task environment, a difficulty metric providing a better link to both the brain and GRU-based machine learning models.

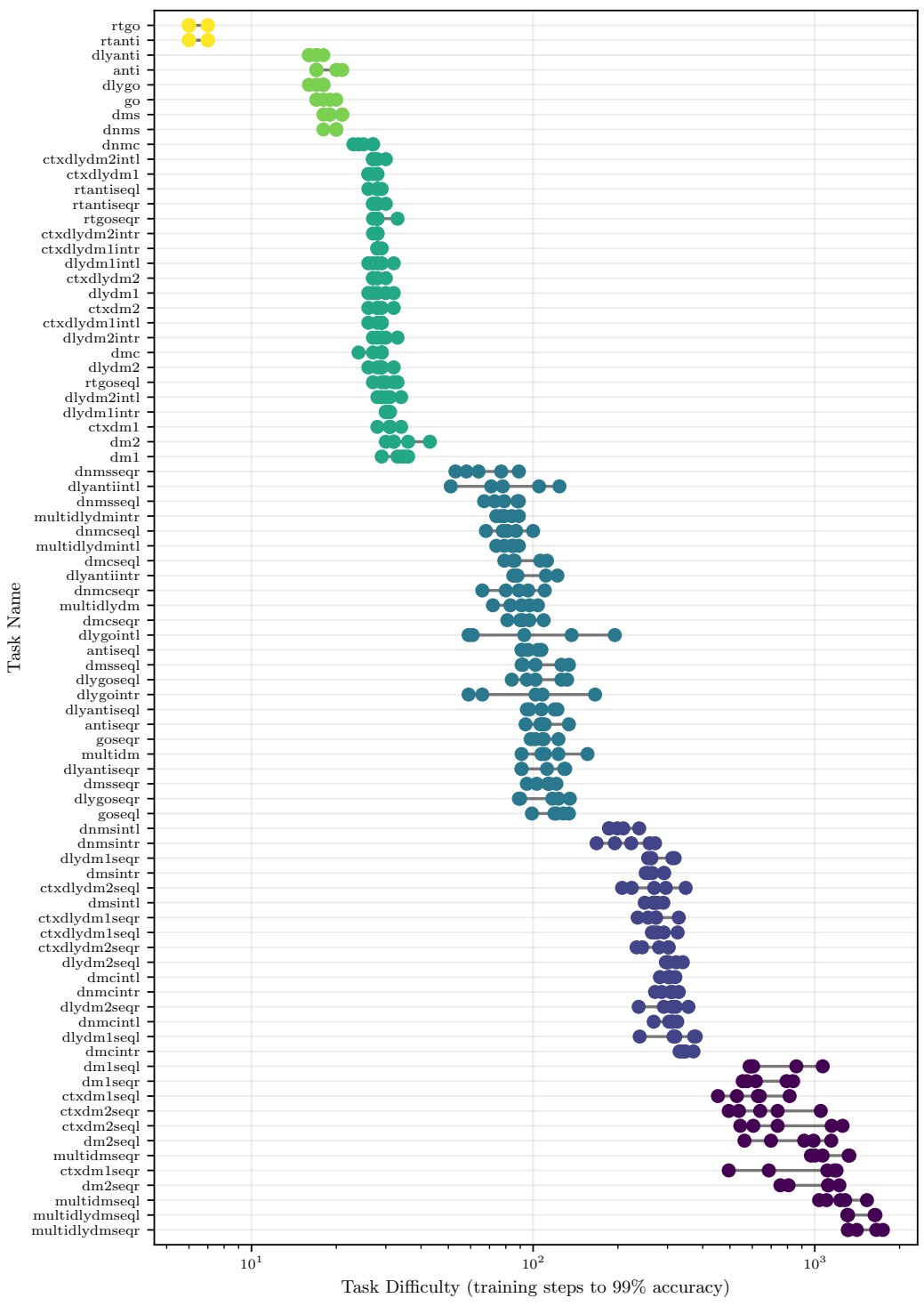

Figure 15: Five measurements of task difficulty made for each task in the Mod-Cog task suite [24]. Tasks are sorted by the median of the five measurements. Interestingly, the tasks form groups around similar difficulty levels, which we indicate with the color of each dot.

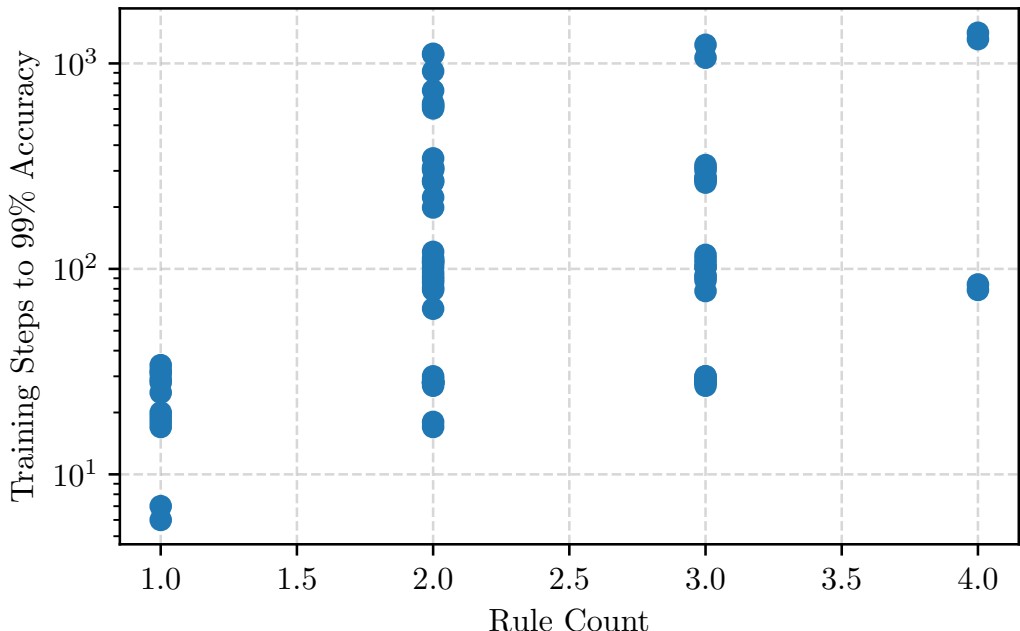

Figure 16: The number of rules that form a Mod-Cog task is correlated with the number of steps it takes a single RNN to learn the task ($r = 0.39$, $p < 0.0001$).

