# OpenReview forum: "Brain-Like Processing Pathways Form in Models With Heterogeneous Experts"
_NeurIPS.cc/2025/Conference — NeurIPS 2025 poster_

### Official Review · Reviewer_VsUx · 2025-06-30

**Clarity:** 3
**Significance:** 2
**Originality:** 3
**Rating:** 4
**Confidence:** 5

**Summary:**

This work proposes a novel regularisation technique to be used on Hetrogenous Mixture of Expert models (HMoEs) which enforces stronger decoupling of experts. When these experts decouple it means that a few experts are necessarily active at any given time, forming a "pathway". The work posits three normative criteria used to define a pathway and show that their model - the Mixture of Pathways Model  (MoP). Importantly, these criteria also correlate to more practical metrics such as the robustness of the model to lesions and the amount of neurone used to perform more difficult tasks. Finally, the behaviour of the MoP is compared to the brain where it is show that the MoP, similar to the brain, starts learning by increasing pathway complexity but then moves over time to using less complex pathways and "subcortical" regions. Similarly, the model is shown to identify task difficulty and route the task appropriately from within the first epoch, similar to the brain. Baselines HMoEs are shown to display neither biological identified behaviours.

**Questions:**

To state my above points as questions:
1. Is counting the number of neurons in an expert a valid proxy for network module complexity?
2. Could the authors elaborate on some of the tasks within the Mod-Cog task set (Go-NoGo is obvious for the target community, but what does one of the more difficult tasks looks like)?
3. Is there even an idea from the literature on how the human brain forms clear pathways and performs the resource allocation similar to section 5 (it's fine if not, I just think if there is one it would help to be explicit)?

**Ethical Concerns:**

["NO or VERY MINOR ethics concerns only"]

**Final Justification:**

My original assessment of the work was positive. My primary two concerns were the weak correlation between neuron counting and module complexity, and the lack of sufficient detail to keep either machine learners or neuroscientists happy.

From the rebuttals the authors agreed to add some detail to the main text, which satisfies the second point to a degree. In response to the first point the authors mentioned an experiment on the GRU experts recurrent weights where they measured the participation as a better correlate of module complexity. The participation ratio of the bigger network was larger (which is to be expected and supports their claims) but this metric is not a measure of rank, rather a measure of sensitivity to input. I do acknowledge that no great measure of a network modules rank exists, but the persistence of the environment counting approach still raises questions for me about the claims.

Overall I think the work can be admitted to the conference and presents results of interest to ML and neuroscience communities (I agree with the arguement in the rebuttal outlining the need for more models of neuroscience principals too). This interdisciplinary nature of the work is overall a strength for me and mitigated the lack of detail in places.

**Limitations:**

Yes and the limitations mentioned show a good deal of insight in their own right.

**Quality:**

3

**Strengths And Weaknesses:**

# Strengths
## Quality
Overall the work is of a high quality. It is well motivated and grounded firstly in the literature. I appreciate the effort put into contextualising the model and honestly discussing its limitations in neuroscience and machine learning. The hypothesis of the work is clear and the normative criteria are motivated quite well. The tasks used are appropriate for the stated goal of the work and results are interpreted fairly and clearly reflect the hypothesis being evaluated. Overall I think the experimental design of the work is well done, again given the goals of the work.

## Clarity
The work is very well written and structured appropriately. Figures are clear with helpful captions. Notation is intuitive and used sparingly. The use of the words "consistent", "self-sufficient" and "distinct" for describing the normative criteria is effective and makes each criteria memorable which aids the clarity of the remainder of the work.

## Significance
This work presents some interesting results which should be broadly applicable to the machine learning and neuroscience communities. The highlight for me is the results showing the sensitivity of HMoEs to deactivations on Lines 211 and 212. This should be a point of concern for all who work on MoEs. Conceptually, the work is important and I think the general discussion linking machine learning and neuroscience should also be considered when evaluating the significance of this work. The work speaks well to both communities and the biologically motivated criteria provide principals which could be foundational to future work in machine learning. I do have some qualifications to this which I discuss below, but taking the work on its own terms and considering the paper as a whole (not just the regularisation terms or results) I think there is a lot here which can inform future work.

## Originality
The work benefits from being inter-disciplinary here as I have not seen many works compare MoEs directly to explicit principals from neuroscience. The additional regularisation terms themselves are also interesting and the combination of different mechanisms is original (while dropout itself is obviously not). Finally, the experimental design is also original, for example by tracking the initial usage of different pathways after one epoch of training and also tracking LPC over time, which is enabled by the original set of criteria this work introduces. Overall the originality is the strongest point of this work.

# Weaknesses
## Quality
My only concern for quality is the exact definition of LPC, specifically, it is completely agnostic to the behaviour of the model itself. As a result, this definition relies heavily on the assumption that an expert, when used, will use all of its neurons for the given computation. Moreover, even if the model did use all of its neurons there is no guarantee that the actual computation being performed is more complex than a smaller module. For example, it has been shown that neural networks trained from small weights have an inductive bias towards low-rank solutions [1] and in the case of linear modules, the rank of the computation being performed depends entirely on the dataset [2]. While I don't think the LPC metric is wrong or unjustified here, I think some clear discussion is necessary about the implicit assumptions linking number of neurons to pathway complexity. I do not think this invalidates the results and as a whole I still find the experimental results compelling. But take the baseline in Figure 5 for example, this behaviour could likely result from wider networks having a higher propensity to better initialised pathways (like a lottery ticket argument [3]) but then these wider networks could still use fewer neurons or low-rank computations. As most results use the LPC, this does limit the quality of the work for me, and I do hope I am missing something which can be clarified in the rebuttal.

## Clarity
My only issue on clarity is that some notation is used before it is introduced or inconsistently. For example, $L_{fix}$ and $L_{response}$ in Algorithm 1. Algorithm 1 is even referenced in the text before this notation is used (so it's more than a placement issue of the text box). Similarly, $m^2$ is not defined on Line 167. On Line 188, $L_{act}$ is being reference but I think this is meant to be $L_{response}$. But these are minor points.

## Significance
Inter-disciplinary work is difficult but crucial, and this should be taken into account when considering the significance of this work. However, this work falls slightly short in speaking to either a machine learning or neuroscience community. For example, the absence of more experiments does limit the applicability of the model to machine learning communities - and even the fact that the benchmarks which are used are left entirely to the appendix, while machine learning communities would like to understand the task setup. On the neuroscience side, relatively little explanation of the potential mechanisms for the criteria in the human brain is given. The biological plausibility of dropout is mentioned and more of these sorts of statements would have helped a lot in grounding the work more clearly in the neuroscience. Similarly, the comparison of simpler and more complex pathway to sub-cortical and cortical regions does not seem to be extremely concrete beyond just the movement of computation between these regions over time. But a similar comparison to the dorsal and ventral striatum would also fit as initial deliberate learning is transitioned into more innate habitual behaviours (I am not saying this makes more sense just saying this comparison would also fit the general description of events). While the experts consist of hidden neurons, making the comparison to cortex more natural than striatum, the work would benefit from a more concrete mapping. However, I reiterate that these complaints are mitigated a lot by the well-done Discussion which pre-empts most of these points and also explains other considerations. It would be impossible to please everyone and I think the work as a whole is very insightful. But it would benefit from a few key points being made to link model computation more explicitly to biological mechanisms and a greater focus on the tasks being performed to inform future work in machine learning.

[1] Huh, Minyoung, et al. "The low-rank simplicity bias in deep networks." arXiv preprint arXiv:2103.10427 (2021).\
[2] Saxe, Andrew M., James L. McClelland, and Surya Ganguli. "Exact solutions to the nonlinear dynamics of learning in deep linear neural networks." arXiv preprint arXiv:1312.6120 (2013).\
[3] Frankle, Jonathan, and Michael Carbin. "The lottery ticket hypothesis: Finding sparse, trainable neural networks." arXiv preprint arXiv:1803.03635 (2018).

---

> ### Author Rebuttal · Authors · 2025-07-30
>
> Dear Reviewer,
>
> Thank you for taking the time to review our paper and sharing your ideas for improvements. We are excited to see that you found our work was well motivated, following a suitable experimental design, with results being interpreted fairly and the report being of high quality. We hope the following addresses your open questions. Please note that we are not able to update the PDF at this stage, but all clarifications will be represented in the camera-ready version of our manuscript.
>
> ## Questions
>
> **(1: Computational complexity of network)** Thank you for bringing up your ideas surrounding the lottery ticket hypothesis (LTH) and low-rank learning. These are very interesting ideas and align well with several follow up research questions we have since been planning internally. We will address the LTH and low-rank issues separately:
>
> *(1.1: Do we only use large experts for complex tasks because the LTH makes it easier to learn complex tasks there?)* We agree that part of the reason why large experts are used is because this makes learning easier and this idea actually aligns with what we show in our paper. As we discuss in Section 5.2 and Figure 6, the brain (and our model) first learn tasks in “overly complex network configurations,” only after which they are gradually moved to simpler networks. Our model’s explanation for this phenomenon is that initially, learning tasks in overly complex pathways helps learning by avoiding local minima of optimizing routing loss before the task loss [ref.1]. Your LTH viewpoint is a related theoretical point, explaining why it is simpler to learn in the complex networks at first, but Figure 6 also shows that the network then works to gradually push tasks downwards towards the correct “minimal network complexity.” While the network complexity chosen during learning is partially reflective of what supports learning, during training tasks are then moved to pathways that better reflect the task’s actual difficulty.
>
> *(1.2: Does the number of neurons capture the actual processing complexity of each expert)* We appreciate your point that large networks can still converge on low-rank solutions, so that by just using the number of neurons, we do not know for sure whether the larger networks are actually using all of their hidden units to implement more complex functions. Your idea of considering the rank of networks was very helpful for running additional analyses on this. Specifically, we analyzed the effective ranks of our GRU experts’ recurrent weight matrices as the participation ratio of the squared sum of singular values to the sum of squared singular values, (Σσᵢ)²/(Σσᵢ²) [ref.2], which measures how evenly distributed the singular values are and thus how many dimensions the matrix effectively uses. We find that all experts across both model types decrease slightly in effective rank over training, but there is no difference between the two models in how they decrease (p>0.05). For both models, we find that the effective rank of large experts is roughly double that of small experts, both before and after training, so that large experts always have a much larger effective rank than small experts (p = 9e-25). We think these additional analyses support the conclusion that in our case the neuron number is at least a good approximation of processing complexity, and we will add these helpful results to our paper.
>
> **(2: Elaborate on details of ModCog tasks)** Figure 7 in the appendix provides an overview of 6 sample tasks. Generally what we see here and also in the task names in general, as listed on the y-axis in Figure 14 (also in the appendix), is that the 82 tasks in ModCog are created based on combinations of task motifs and task rules. As an example, Figure 7 shows that the “Delayed Decision Making” task (`dlydm1`) is an altered version of the standard “Decision Making” task (`dm1`) with the additional “Delay” (`dly`) rule. One part of what makes a task difficult is the number of rules that make up a task: we find that the number of rules contained in a task correlates well with our ‘GRU learning time’ task difficulty metric ($r=0.39$; $p<0.01$). This is reflected in Figure 14, which shows that the two most difficult tasks are `multidlydmseql` and `multidlydmseqr`. These are decision making tasks, like `dm1` in Figure 7, with three additional rules. These additional rules are `dly` (delay), `multi` (integration across two stimuli), and `seql` or `seqr` (the response needs to increase monotonically in a specific direction; left or right). We are not allowed to include additional visualizations as part of the review, but the official ModCog GitHub contains some visualizations of this if you would like to see them [ref.3]. Nonetheless, we will include this additional context in the camera-ready version of our manuscript.
>
> **(3: More detailed discussion of related neuroscience findings)**
>
> *(3.1: Resource allocation in the brain)* As we briefly mention in line 354, we would expect the routing mechanism to be implemented by thalamic nuclei [ref.4]. If we looked for a mechanism influencing the preference for simple or complex processing, then the most supported related finding would be “cognitive effort,” akin to “processing power,” which is controlled by the release of norepinephrine in the locus coeruleus [ref.5, ref.6, ref.7]. In our model, this “cognitive effort” would translate to the increased use of complex experts. The locus coeruleus projects to thalamic nuclei, presenting a means for influencing the routing to simple or complex processing pathways. An additional mechanism could be found in the hypothalamic hypocretin/orexin neurons which play a specific role in metabolic control and might govern the amount of processing power as a function of metabolic budgets [ref.8]. These hypothalamic neurons similarly have strong projections to thalamic neurons and could also influence routing. We will include this context in our final paper.
>
> *(3.2: Pathways formation in the brain)* Less empirical research exists on how pathways form to begin with. This is partially due to how they develop both within lifetime but also over evolution, making it difficult to study empirically, and making it especially important as a topic for theoretical investigation. We see this as a strong motivation for our work. One factor behind pathways and network formation that has been found across studies is the influence of both spatial metabolic costs and communication optimisation [ref.9, ref.10]. More recent investigations have shown that these metabolic constraints need to be studied alongside functional (task) optimisation to observe how modules and pathways form over time [ref.11, ref.12], suggesting strong interaction effects between structural and functional optimisation goals. We briefly discuss these principles in lines 60-63 and 174-177. The limitations of this existing work come from the fact that they (a) do not have heterogeneous submodules / regions in their models and (b) do not allow for dynamic combination of modules, but instead have static pathways, both of which are likely unrepresentative of the brain’s systems-level architecture [ref.13]. In our work, we extend the idea that pathway formation might be driven by metabolic optimisation (“avoiding complex processing where possible”), but bring it to a new type of model that allows us to capture the brain’s systems-level architecture more truthfully. With this, we were then able to identify the more specific inductive biases (beyond just metabolic cost), and show how dynamics between cortical and subcortical processing / learning arise in this new architecture.
>
> **(4: Additional point from clarity section / clarification on notation)** Thank you for pointing out these errors. $m^2$ should be written as $s_j^2$, and $L_{act,i}$ should be written as $L_{response,i}$ as you said.
>
> We want to thank you again for taking the time to review our work and suggesting improvements to our manuscript. We hope our comments above could clarify all the open questions you had.
>
> ## References
>
> [1] A Simple General Approach to Balance Task Difficulty in Multi-Task Learning: https://arxiv.org/abs/2002.04792
>
> [2] A scale-dependent measure of system dimensionality: https://www.sciencedirect.com/science/article/pii/S266638992200160X
>
> [3] Mod-Cog tasks for multitask learning: https://github.com/mikailkhona/Mod_Cog
>
> [4] The role of the thalamus in the flow of information to the cortex https://royalsocietypublishing.org/doi/10.1098/rstb.2002.1161
>
> [5] Cognitive effort: A neuroeconomic approach: https://link.springer.com/article/10.3758/s13415-015-0334-y
>
> [6] AN INTEGRATIVE THEORY OF LOCUS COERULEUS-NOREPINEPHRINE FUNCTION: Adaptive Gain and Optimal Performance: https://www.annualreviews.org/content/journals/10.1146/annurev.neuro.28.061604.135709
>
> [7] Orienting and Reorienting: The Locus Coeruleus Mediates Cognition through Arousal: https://www.sciencedirect.com/science/article/pii/S0896627312008197
>
> [8] Neurometabolic signaling and control of policy complexity: https://www.biorxiv.org/content/10.1101/2025.02.24.639890v1
>
> [9] Simple models of human brain functional networks: https://www.pnas.org/doi/abs/10.1073/pnas.1111738109
>
> [10] A generative network model of neurodevelopmental diversity in structural brain organization https://www.nature.com/articles/s41467-021-24430-z
>
> [11] A single computational objective drives specialization of streams in visual cortex https://www.biorxiv.org/content/10.1101/2023.12.19.572460v1
>
> [12] Spatially embedded recurrent neural networks reveal widespread links between structural and functional neuroscience findings https://www.nature.com/articles/s42256-023-00748-9
>
> [13] Building artificial neural circuits for domain-general cognition: a primer on brain-inspired systems-level architecture https://arxiv.org/abs/2303.13651

---

> > ### Comment · Reviewer_VsUx · 2025-08-08
> > **Response to Authors**
> >
> > I thank the authors for their detailed response. Overall I think my concerns are fairly well addressed - contingent on the promised changes being made.
> >
> > The experiment with the participation ratio is useful but somewhat flawed in its own right (why only consider the recurrent pathways?). I will reiterate that I don't think counting the neurons is a good approach at all and so the participation ratio would need to be used through the work as a more helpful metric.
> >
> > For the comments on more information being needed about the tasks being conducted and the neuroscience links. Thank you for the added details. My feeling is that these should have been in the main text - I am aware that the ModCog visualisations can be found on their GitHub and that you had some elaborations in the appendix. Space is limited I know, but some no mention of the tasks or allusions to thalamic nuclei didn't feel like enough.
> >
> > Overall I still feel like I am sitting on a weak accept since the neuron counting metrics are quite embedded in the paper and it's messaging (but overall I am very happy for this paper to be presented at the conference) but in recognition of the proposed changes I will raise my clarity score and confidence.

---

> ### Author Response · Authors · 2025-08-08
>
> Dear Reviewer,
>
> Thank you for your comments and clarifications on your thoughts. We briefly wanted to highlight that we agree that further details regarding the task and neuroscience links will be valuable additions to the main text, and we will make sure that the additional page we receive for the camera-ready version will elaborate on these details, as well as other analyses that were added during the rebuttal. We also want to quickly clarify that if we used the participation ratio of the recurrent matrix instead of the neuron number, this would not impact the results, as the relative participation ratio for the recurrent weights when compared between networks stays pretty much constant throughout training. We would be happy to run additional tests on using other weight matrices for the camera-ready version.
>
> Thank you again for taking the time to review our manuscript and providing helpful comments for improving our work.

---

### Official Review · Reviewer_gQat · 2025-07-02

**Clarity:** 3
**Significance:** 2
**Originality:** 3
**Rating:** 5
**Confidence:** 4

**Summary:**

This work explores the formation of processing pathways in mixture-of-experts (MoE) models. The heterogeneous MoE models studied consist of three layers, each with three experts, of which one is an identity connection, one is smaller, and one is larger. The authors first establish several improvements over their baseline to stabilize routing and improve their self-sufficiency. The improved model behaves in a more brain-like fashion, in that it solves simpler tasks without the complex expert more often, and it allocates the complex expert to more complex tasks early in training. These findings suggest that a simple mixture of heterogeneous experts, by itself, does not explain the formation of heterogeneous pathways in the brain, and suggest a potential improvement in MoE models in ML.

**Questions:**

- See also the limitation below – the work appears predicated on simulations with a single model configuration. Could you support the lack of any ablations, and/or better motivate the specific hyperparameters chosen?
- In the introduction of the model in Algorithm 1, the routing weights appear to be indexed by the layer and expert, $w_{l,j}$. Later, in the definition of the _Learned Pathway Complexity_ in 4.1, they appear to be indexed by task, timestep, and expert, $w_{i,j,t}$. If my understanding so far is correct, then (1) what happened to the layer indexing in the LPC definition, and (2) perhaps the task could be a superscript, rather than a subscript? My motivation for asking is that I currently don’t understand where layers fit into the LPC definition, as it is currently described.
- Could you motivate the evaluation with dropout in section 4.2? With standard (unit-wise) dropout, the standard is to downweigh all units by the probability of dropout, rather than continuing to drop out in evaluation. Did you try something of this sort?

**Ethical Concerns:**

["NO or VERY MINOR ethics concerns only"]

**Final Justification:**

With the added evaluations, ablations, and clarifications, I think this would make a worthwhile contribution to the proceedings, and I updated my score accordingly.

**Limitations:**

The authors addressed several limitations. However, the authors conduct their experiments using a single, small model configuration. It is hard to evaluate to what extent hyperparameter choices (3 layers, 3 experts, expert sizes) influence the findings. This limitation is important because the work is primarily empirical and simulation-driven.

**Quality:**

3

**Strengths And Weaknesses:**

**Strengths**: The authors study a problem of interest to both neuroscientists and machine learning in a well-motivated setting, using a simple model of a much more complex phenomenon. The improvements the authors build on their baseline model follow well from the limitations identified, and the analyses are presented well.

**Weaknesses:** The entire set of results is predicated on a set of simulations performed with a single hyperparameter setting that is neither particularly strongly motivated nor ablated. As such, it is hard to disentangle to what extent some of these specific choices (enumerated below in the limitations) drive the observed results, vs. to what extent they are broader features of the class of models studied. Even minor ablations would go a long way in strengthening the work.

Other minor notes/questions for the authors:
- L132: Would it be correct to say that the input is a $128 \times 350 \times 115$ input tensor, accounting for the dimensionality of the input? If not, where does the 115 fit in the 128 by 350?
- L181: When incorporating the LPC into the model loss, wouldn’t it suffice to add a weight to the LPC (the role $\alpha$ appears to play)? Could you motivate dividing by the task loss, which would explore as the response loss goes to zero?
- L188: Continuing from the previous question, this line suggests that the normalization is by $L_{act,i}$, but the equation has the normalization by the response loss. Could you clarify?
- L265: If some of this analysis is predicated on the difference between the distribution of cluster assignments, (a) to what extent is it sensitive to the initialization of the K-means clustering procedure, and (b) did you conduct any sort of significance testing to validate that the differences are meaningful?

---

> ### Author Rebuttal · Authors · 2025-07-30
>
> Dear Reviewer,
>
> Thank you for taking the time to review our paper, for sharing your suggestions, and for your attention to detail. We are glad that you appreciated how our work tackles a complex phenomenon relevant to both neuroscience and ML, and we hope the following can address your concerns and open questions. Please note that we are not able to update the PDF at this stage, but all clarifications will be represented in the camera-ready version of the manuscript.
>
> ## Weaknesses
>
> **(1: Ablations)** We have now run additional ablation experiments, varying the alpha parameter, dropout, router size, and embedding layer. We highlight the effects of these ablations through the findings shown in Fig.5 and Fig.6, based on 10 randomly initialized models for each configuration:
>
> |  | Accuracy | Fig.5 correlation | Fig.5 p-value | Fig.6b correlation | Fig.6b p-value |
> |---|---|---|---|---|---|
> | Baseline | **91.03%** | -0.01 | 0.6576 | -0.49 | 0.0000 |
> | Our Model ($\alpha$ = 1e-5; Router dim = 64) | 82.95% | 0.54 | 0.0000 | 0.31 | 0.0039 |
> | No dropout | 90.29% | 0.55 | 0.0000 | 0.03 | 0.7815 |
> | $\alpha$ = 1e-4 | 67.51% | -0.57 | 0.0000 | -0.37 | 0.0004 |
> | $\alpha$ = 1e-6 | 90.00% | **0.62** | 0.0000 | 0.18 | 0.1157 |
> | No task embedding | 82.67% | 0.58 | 0.0000 | **0.58** | 0.0000 |
> | Router dim = 32 | 83.25% | 0.46 | 0.0000 | 0.33 | 0.0023 |
> | Router dim = 128 | 82.17% | 0.35 | 0.0013 | 0.27 | 0.0125 |
>
> These new ablations do show some interesting new effects. Generally we observe that our model lies at a good intersection of accuracy and strong correlations for Fig.5 and Fig.6, indicating the formation of brain-like pathways. The router dimension does not seem to strongly affect the overall pattern. As expected, selecting the correct alpha value is crucial: if it is too high, the model gives up on learning complex tasks and does not properly converge. If it is too low, the model does not have enough pressure to assign tasks to the most simplest pathway possible, leading to the weak Fig.6 correlation. Having no dropout naturally helps task performance, but it also has the interesting effect of causing the Fig.6 correlation to disappear. This suggests that brain-like learning, as shown in Fig.6, develops as an interaction of the LPC loss and the dropout. Lastly, we find that the model without the embedding layer shows the same effects as our main model, with even stronger correlation. However, including the embedding layer increases training speed by about 250% and was hence helpful for us to conduct our experiments. We will include these additional results and analysis in the camera-ready version of our manuscript.
>
> **(2: Dimensionality of input tensor; L132)** Thank you for pointing this out, the dimensionality of the input tensor is 128x350x115. We will update this in the text.
>
> **(3: Incorporating LPC into the model loss; L181)** You are correct, the loss can be written in a simplified form as $L=L_{fix}+\sum_i^\mathcal{T}(L_{response,i}+\frac{\alpha LPC_i}{L_{response,i}+\epsilon})$. This changes the equation slightly, specifically $\epsilon$ would need to become $\frac{\epsilon}{T_i}$ in order to be equivalent. However, in practice, $\epsilon$ is not sensitive to changes, as it is a very small value used simply to avoid division-by-zero errors. We will verify that it does not change our results and plan to update Equation 2 in the camera-ready version to improve clarity.
>
> **(4: Motivate division by task loss; L181)** This can be motivated from the perspective of both ML and neuroscience. *On the ML side*, it is an established phenomenon that when a loss is made up of two or more parts, one of which is significantly easier to minimize than the other, that this can result in optimization issues such as convergence on local minima, as studied in the context of multi-task optimization [ref.1]. In our case, one such local minimum would involve our model leveraging the simplest pathways possible, reducing the routing loss to zero, while failing to solve most or all tasks correctly. We found that normalization helped us avoid this. *On the neuroscience side*, we know that “cognitive effort,” or “processing power,” is affected by norepinephrine release from the locus coeruleus in the brainstem [ref.2, ref.3]. In our model, this would correspond with usage of the complex expert in each layer. It has been observed that activity in the locus coeruleus peaks during early stages of learning when there are large error signals, and weans as skills are acquired [ref.4]. Normalizing our routing loss by the task performance achieves this same effect. We will add this additional context to our paper in Section 4.1.
>
> **(5: Clarify division term; L188)** Thank you for bringing this error to our attention, $L_{act,i}$ should be $L_{response,i}$. This will be corrected in the text.
>
> **(6: Dependence of cluster assignment on K means parameter)** Thank you for pointing this out, we have now run tests with 10 different K-means initializations. In the original manuscript, we observe that our model tends to develop a couple of dominant routing patterns used for many tasks at the same time, as shown by the longer-tailed distribution. To statistically test for a difference between the clustering patterns of our model and the baseline model we measured the size of the largest observed cluster for each model and found that our model developed more dominant large clusters ($p=0.014$; see line 270). With the new additional tests included, this p-value is further reduced to $p<0.0001$.
>
> ## Questions
>
> **(1: Ablations)** We outline our new ablations above.
>
> **(2: Indexing when Algorithm 1 is introduced)** Thank you for spotting this inconsistency. We added the $l$ subscript in Algorithm 1 since this algorithm needs to explain how $w$ is calculated, whereas Equations 1 and 2 simply assume it exists, but in hindsight this is confusing. We will remove the layer index from Algorithm 1, and move the expert index $j$ to be a superscript in all mentions of $w$. We hope this makes sense, but welcome further suggestions on this point.
>
> **(3: Evaluation with dropout)** We do not enable expert dropout during evaluation, following typical procedure for evaluating standard dropout layers [ref.5]. The one exception to this is Figure 3, where the “evaluation accuracy” is calculated with dropout enabled according to the threshold shown on the y-axis. We now also provide an additional ablation without any dropout used during training, as shown in our first point of this rebuttal. We have not experimented with other forms of dropout.
>
> We want to thank you again for taking the time to review our work and suggesting improvements to our manuscript. We hope that our comments above clarify all the open questions you had.
>
> ## References
>
> [1] A Simple General Approach to Balance Task Difficulty in Multi-Task Learning: https://arxiv.org/abs/2002.04792
>
> [2] Cognitive effort: A neuroeconomic approach: https://link.springer.com/article/10.3758/s13415-015-0334-y
>
> [3] AN INTEGRATIVE THEORY OF LOCUS COERULEUS-NOREPINEPHRINE FUNCTION: Adaptive Gain and Optimal Performance: https://www.annualreviews.org/content/journals/10.1146/annurev.neuro.28.061604.135709
>
> [4] Orienting and Reorienting: The Locus Coeruleus Mediates Cognition through Arousal: https://www.sciencedirect.com/science/article/pii/S0896627312008197
>
> [5] Pytorch documentation on dropout: https://docs.pytorch.org/docs/stable/generated/torch.nn.Dropout.html

---

> > ### Comment · Reviewer_gQat · 2025-08-05
> >
> > Thank you for your thorough rebuttal. I have no additional questions and used your responses to inform my final assessment.

---

### Official Review · Reviewer_A6SL · 2025-07-02

**Clarity:** 2
**Significance:** 3
**Originality:** 3
**Rating:** 3
**Confidence:** 3

**Summary:**

This paper studies when MoE architectures produce patterns of expert activation that match certain properties of information routing in the brain. The main findings are that regularizing with a task-dependent measure of expert usage leads that measure to correlate across independent runs of the model and that training with expert dropout makes the models more robust to dropout at test. These interventions also yielded more long-tailed distributions when clustering tasks by their usage patterns.

**Questions:**

How much do your results depend on details of the router? Inductive bias for sparsity seems especially relevant.

The denominator in eq 2 doesn’t match the text description, since L_response,i is negatively related to the model’s ability to solve task i. This discrepancy may be related to the undefined L_act,i in the text.

Sec 4.2: please state which interventions are done during training and which during evaluation.

Sec 4.3, fig 4: Your model clearly shows more structure within each cluster across phases and expert types compared to baseline (this follows unsurprisingly from the regularization in eq 2), but what is the evidence for the claim that it produces more-distinct routing patterns? Within both models, all the clusters seem to show nearly the same pattern.

**Ethical Concerns:**

["NO or VERY MINOR ethics concerns only"]

**Final Justification:**

The paper offers a novel way of thinking about brain-like information routing in NNs but falls short in several ways. The notion of pathway is too vague. The primary results are built in by the regularization scheme which encourages the model to use certain experts for certain tasks. This also makes the identified pathways only correlational; there's no evidence of synergy or coordination between experts.

**Limitations:**

yes

**Quality:**

3

**Strengths And Weaknesses:**

Strengths:

Clever and novel ways of thinking about brain-like information routing in NNs

Diverse set of experimental evaluations

Weaknesses:

The notion of pathways needs to be more carefully defined and better justified. I didn’t understand what you meant until eq 1 (even sec 4.0 was vague and unclear). FWIW my guess until sec 4.1 was that pathways meant the experts form paths or chains: high weighting for some expert at one layer is consistently (within a model and across tasks) followed by high weighting for some other expert at the next layer.

I’m also not seeing the motivation for why pathways in your sense is a desirable emergent property.

Using the same LPC as both a regularizer and a metric biases the results. The task-specific scaling in eq 2 is especially problematic: if the model is explicitly penalized for LPC more on some tasks than others then of course it will consistently produce lower LPC for those tasks.

Likewise the correlation in fig 5 is unsurprising since the model was explicitly penalized for LPC on easier tasks.

The measure of task difficulty used (training steps needed by a standalone GRU) is too crude for comparison to brain patterns. What matters for patterns of brain activation is not raw difficulty (which can be high for simple tasks like detection in strong noise) but complexity of rules or working memory demands.

---

> ### Author Rebuttal · Authors · 2025-07-30
>
> Dear Reviewer,
>
> Thank you for taking the time to review our paper! We are glad that you found our model a clever and novel way of characterizing brain-like information routing in neural networks. We hope the following addresses your concerns and questions. Please note that we are not able to update the PDF at this stage, but all clarifications will be represented in the camera-ready version of the manuscript.
>
> ## Weaknesses
>
> **(1: What are pathways)** Thank you for pointing out that this was unclear. We think of pathways as groups of experts that tend to be used in tandem, where being part of one pathway does not prevent an expert from also being part of another pathway. In that sense, they are patterns of “recurring co-usage” that are dynamically activated across tasks and time periods within tasks.
>
> **(2: Why are pathways desirable)** From a neuroscience point of view, we argue that this “recurring co-usage” of regions is what takes place in the brain. There are dominant patterns in which brain regions tend to be co-activated, but any given brain region can be part of multiple different subnetworks, akin to neural reuse [ref.1]. From a large-scale AI point of view, we know that MoE models are parameter efficient, but the ad-hoc loading of experts can limit the speed of execution of these models. Predictable routing has been shown to help that [ref.2; see line 82], and we think pathways provide a good balance between diverse expert usage and predictable routing patterns in a way that resonates with the functional architecture of the brain.
>
> **(3: Bias through using LPC in regulariser and analyses)** We want to address this point using results from three different perspectives: two from our existing manuscript, and one from new investigations we include in the rebuttal.
>
> *(3.1: In relation to Figure 5)* We agree that this specific finding follows logically from the setup of our model once it has been shown to work. However, the mechanisms underlying pathway formation between heterogeneous regions were unknown beforehand: another reasonable hypothesis could have been that the baseline architecture (layers of heterogeneous experts without any inductive biases) learns to allocate computation toward more difficult tasks automatically. Figure 5 shows that this is not the case unless our specific inductive biases are included in the model architecture. We think one of the key benefits of our model is that it delivers an understandable implementation for a set of complex phenomena relating to how pathways form and how they behave in a brain-like fashion.
>
> *(3.2: In relation to other findings in the paper)* Like you said, penalizing the model with the LPC incentivizes it to reduce its LPC for each task. However, if routing decisions were only based on a task’s LPC, we might expect that routing is constant within each trial of a given task. Instead, our model generates nuanced routing dynamics across timescales within single trials of tasks, as can be seen in Figure 12. It discovers that for some groups of tasks, it can use a generalizable routing pattern, while others require a very unique usage of experts. This is shown through clusters on the y-axis of Figure 12 and discussed in Section 4.3 (please note our response to your related question below). These findings suggest that the model has learned a general strategy of allocating computational resources across the 82 tasks. With the model and its overall principles established through this paper, future work can now deliver more detailed investigations on how far this extends to completely new tasks.
>
> *(3.3: New findings from rebuttal in relation to Figure 6)* Motivated by your concerns, we ran some additional experiments that analyze specific aspects of our model (dropout, routing) on LPC and pathway formation. Notably, we found that when trained with our normalized LPC but without dropout, our model exhibits the effect shown in Figure 5, but not the effect shown in Figure 6. This indicates that the finding in Figure 6 is due to an interaction between LPC regularization and dropout, and is not explained by training with LPC regularization alone. We speculate that dropout forces the model to be explicit about which pathway is used in learning and is crucial for creating brain-like learning dynamics.
>
> **(4: Oversimplified task difficulty metric)** There are many ways to characterize the difficulty of learning problems [ref.3], but no universal complexity measure has been developed to date. We believe “training steps needed to learn the task” is a sensible measure because this can be measured without any implicit biases and takes into account task demands such as working memory and any other factor making inference problems challenging [ref.3]. At the same time, we agree it is slightly unclear whether such a complexity measure would neatly map onto what humans perceive to be “difficult tasks,” which is often linked to the number of rules in a task [ref.4], as you suggested. We find that our difficulty metric is in fact correlated with the number of rules in each task (r=0.39; p<0.0001), and that our model exhibits an even stronger correlation for the brain-like finding in Figure 5 when this is used as the difficulty metric (r=0.57, p<0.0001) compared to the baseline model (r=-0.09, p=0.4288). At the same time, using “number of rules per task” as the actual difficulty metric has two downsides: (a) it is a discrete and ordinal measurement from 1 to 4 with less statistical power, and (b) some rules are harder to learn than others (i.e. go vs. dm1). This suggests to us that our current convergence-based complexity measure is, at least in this specific task environment, a difficulty metric providing a valid link to both the brain and GRU-based ML models.
>
> ## Questions
>
> **(1: Effect of details of router on results)** We have now run some additional experiments to better understand certain aspects of the router on our model's performance. We highlight the effect of these ablations through extracting the correlations shown in Fig.5 and Fig.6, based on 10 models run for each configuration:
>
> |  | Mean task accuracy | Fig.5 correlation | Fig.5 p-value | Fig.6b correlation | Fig.6b p-value |
> |---|---|---|---|---|---|
> | Baseline | **91.05%** | -0.01 | 0.6576 | -0.49 | 0.0000 |
> | Our Model ($\alpha$ = 1e-5; Router dim = 64) | 82.97% | 0.54 | 0.0000 | 0.31 | 0.0039 |
> | No dropout | 90.10% | 0.55 | 0.0000 | 0.03 | 0.7815 |
> | $\alpha$ = 1e-4 | 69.02% | -0.57 | 0.0000 | -0.37 | 0.0004 |
> | $\alpha$ = 1e-6 | 89.68% | **0.62** | 0.0000 | 0.18 | 0.1157 |
> | Router dim = 32 | 83.15% | 0.46 | 0.0000 | **0.33** | 0.0023 |
> | Router dim = 128 | 81.95% | 0.35 | 0.0013 | 0.27 | 0.0125 |
>
> We observe that our model lies at an intersection of good accuracy and strong significant correlations for our Fig.5 and Fig.6 findings. The router dimension does not seem to strongly affect the overall pattern. As expected, the correct alpha value is critical: setting it too high leads to an over-emphasis on routing cost, causing the model to give up on learning complex tasks and fail to converge properly. Smaller alpha values increase accuracy, but form weaker pathways, as there is not enough pressure to assign tasks to the simplest possible pathways. Having no dropout naturally helps task performance, as the model now gets to use all experts at all times, but it also has the interesting effect of causing the Fig.6 correlation to disappear. This suggests that the findings of brain-like learning as shown in Fig.6 develops as an interaction of the LPC loss and the dropout. We will include these results and analysis in our final manuscript.
>
> **(2: Denominator in eq.2)** Thank you for spotting this error, which we will correct. $L_{act,i}$ should be written as $L_{response,i}$ in the text, and we will update line 182 to clarify that we are dividing by the model’s loss, not accuracy, on task i.
>
> **(3: Clarify interventions during inference and training)** We apologize for the confusion regarding interventions during evaluation. Dropout is used during training but not evaluation, aligned with what is typically done in ML. The one exception to this is Figure 3, where the “evaluation accuracy” is calculated with dropout enforced according to the threshold given on the y-axis in order to study our “self-sufficiency of pathways” criterion.
>
> **(4: Evidence for more distinct clusters; Sec 4.3, fig 4)** Thank you for flagging this issue. Upon inspection, we noticed an issue with how we generated Figure 4, making our clustered pathways appear much more similar than they actually are. Unfortunately we are unable to share new figures during the rebuttal, but luckily Figure 12 and Figure 13 show more detailed (and correct) routing patterns. While in Figure 13 (baseline) the routing patterns seem very random and unstructured, Figure 12 shows much more structured routing with distinct pathways. For example, notice how clusters 2 and 3 (labeled on the left side of the figure) rely more on increased processing complexity during the response period, while clusters 5 and 6 do so during the prestimulus and stimulus phase.
>
> We want to thank you again for taking the time to review our work and suggesting improvements to our manuscript. We hope our comments above were able to address the issues you raised in your review.
>
> ## References
>
> [1] Neural reuse: A fundamental organizational principle of the brain: https://pubmed.ncbi.nlm.nih.gov/20964882/
>
> [2] Expertflow: Optimized expert activation and token allocation for efficient mixture-of-experts inference: https://arxiv.org/abs/2410.17954
>
> [3] How Complex is your classification problem? A survey on measuring classification complexity: https://arxiv.org/abs/1808.03591
>
> [4] Goal Neglect and Spearman’s g: Competing Parts of a Complex Task: https://pubmed.ncbi.nlm.nih.gov/18248133/

---

> > ### Comment · Reviewer_A6SL · 2025-08-07
> >
> > Thanks for the thorough and thoughtful replies. They resolve many of my previous comments.
> >
> > On the other hand, the concept of pathways still seems poorly conceived or incoherent. The paper talks about pathways *between* layers, implying the experts are working together, passing information to perform some coherent computation. However, the results don’t bear on this question because they’re only correlational. To really demonstrate pathways I think you would need some analysis of dependency among experts. For example, perturb an expert at one layer and measure how the state or output of an expert at a later layer changes. Pathways should be apparent as sparsity in the resulting expert-expert matrix, as well as consistency of the sparse pattern across tasks.
> >
> > You say in your reply that you shed light on the mechanisms of pathway formation but I don’t see that. You added a regularizer that directly encouraged your correlation notion of pathways and they appeared. Setting aside my previous comments that this is not a surprising result, you didn’t look under the hood to study mechanism. I agree there’s a contribution in showing the baseline model didn’t automatically allocate harder tasks to bigger experts. Fig 6b Baseline may help answer this, as you explain nicely in the paper: the model starts out in the reverse configuration because it punts on the harder tasks.
> >
> > I’m also still not convinced by the ablation experiment (fig 3). A pathway could compute a coherent piece of a task that still isn’t the entire solution, so I don’t see why we should expect the model to give the correct answer with the pathway alone. (The expert dropout is interesting on its own; I just don’t see the connection to the topic of the paper.)

---

> > > ### Author Response · Authors · 2025-08-08
> > >
> > > Dear Reviewer,
> > >
> > > Thank you for taking the time to comment on our rebuttal to further clarify your thoughts. We wanted to elaborate on the points you raise:
> > >
> > > **Pathway definition:** We agree with your assessment that our analysis of pathways currently largely relies on correlational analyses, though we also make use of lesion experiments, as highlighted in the colour scale on Figure 5, where lesioning complex experts specifically affected task accuracy on complex tasks. This approach mirrors neuroscience methods that use lesion data [1] and activation correlations [2,3] to identify networks.
> > >
> > > **Analysis of dependency amongst experts for the definition of pathways:** We agree that this would be a very interesting expansion to gain a deeper understanding of how experts across layers interact to implement a joint computation as a pathway. The task set does allow for an extensive analysis like this to be run, to see how subparts of tasks may be allocated to specific experts [4] and we agree that this would be insightful. Unfortunately, the limited time scope until the end of the rebuttal period does not allow us to run this new analysis, but we believe this is a great opportunity for an independent follow-up investigation.
> > >
> > > **Pathways ought to be apparent as sparsity in the expert-by-expert matrix:** We show matrices like this in Figure 12 and Figure 13 for the pathway model and the baseline model respectively. While in Figure 13 (baseline) the routing patterns are very random and unstructured, Figure 12 shows much more structured routing with distinct pathways. For example, notice how clusters 2 and 3 (labeled on the left side of the figure) rely more on increased processing complexity during the response period, while clusters 5 and 6 do so during the prestimulus and stimulus phase. It is also evident that the usage of experts for the pathway model is sparser than for the baseline model.
> > >
> > > **Mechanism of pathway formation:** We agree that our model provides the opportunity to analyze the exact mechanism of pathway formation in more detail. As you suggest, Figure 6 already points to some deeper mechanistic reasons, specifically that the gradient surface may provide a clearer ‘momentum path’ for learning tasks in complex experts first before gradually pushing them downwards to the simplest possible pathway. As suggested by reviewer VsUx this might be due to the lottery ticket hypothesis [5]. We were not able to fit additional gradient surface analyses into the manuscript but agree that our model provides interesting opportunities to dig into the mechanistic reasons for pathways to develop.
> > >
> > > **Should a pathway be able to provide the entire solution to a problem by itself:** This is an interesting point and links both to your point about dependencies between experts above, but also to a question about what spatiotemporal resolution should a pathway be studied at. When defining our criteria of ‘pathway self-sufficiency’, we took a task-centric point of view. We did so because it was then possible to test whether task accuracy was affected by lesioning experts which were not strongly active during solving this specific task, without any strong additional assumption. It is possible that what we call a pathway, under some conditions, may constitute a combination of other pathways that interact to solve a compositional task, where each pathway focuses on a specific aspect of the task and they jointly solve the overall task. In fact, Figure 12 shows this, as we see how across the different time phases of the task different distinct patterns activate. The current resolution of our analyses stays on the higher-abstraction identifying pathways from a task-centric point of view. However, our model does indeed show much richer patterns which we hope can become the focus of follow up work.
> > >
> > > We want to thank you again for taking the time to review and comment on our work. We hope the above gives some more detail on how we think about pathways in our model at this current point in time, but also how this model can provide the basis for several interesting directions of follow up work.
> > >
> > > **References**
> > >
> > > [1] Fluid intelligence is supported by the multiple-demand system not the language system https://www.nature.com/articles/s41562-017-0282-3
> > >
> > > [2] Network variants are similar between task and rest states https://doi.org/10.1016/j.neuroimage.2021.117743
> > >
> > > [3] Broad domain generality in focal regions of frontal and parietal cortex https://www.pnas.org/doi/10.1073/pnas.1315235110
> > >
> > > [4] Flexible multitask computation in recurrent networks utilizes shared dynamical motifs  https://www.nature.com/articles/s41593-024-01668-6
> > >
> > > [5] The Lottery Ticket Hypothesis:  https://arxiv.org/abs/1803.03635

---

### Official Review · Reviewer_WDiN · 2025-07-04

**Clarity:** 3
**Significance:** 3
**Originality:** 3
**Rating:** 5
**Confidence:** 3

**Summary:**

The paper studies how task specific processing pathways form in neural nets with mixture of heterogeneous experts. Inspired by brain’s pathways among regions, the work extends the heterogeneous MoE architecture to observe conditions in which pathways form. The authors identify three inductive biases that lead to formation of such pathways:

1. A routing complexity cost to penalize use of larger experts
2. Scaling routing loss based on the task complexity, reducing the penalty for hard tasks
3. Stochastic expert dropout during training which leads to self sufficiency

They train models on the Mod-Cog suite of 82 timeseries cognitive tasks.
Authors introduce the learned pathway complexity metric to quantify expert usage per task. Correlating LPC across 20 seeds and clustering tasks that use similar pathways, they show that baseline HMoEs do not form stable pathways (mean LPC correlation ≈0.03).
Adding routing cost and scaling shows high consistency (mean LPC correlation ≈0.71). Expert dropout retains ≳75% accuracy when pruned, showing pathways become more self sufficient.

The resulting pathways behavior is inline with known brain dynamics (cortical vs. subcortical engagement for hard vs. easy tasks).

**Questions:**

1. Can this approach be studied for tasks other than timeseries?
2. Can authors provide a per task accuracy table to show the effect of routing cost or dropout per task?
3. Do different architectural choices in Appendix A.1 change pathway formations in any form?

**Ethical Concerns:**

["NO or VERY MINOR ethics concerns only"]

**Final Justification:**

Authors have addressed my concerns and I continue to support acceptance of this work.

**Limitations:**

Yes

**Quality:**

3

**Strengths And Weaknesses:**

strengths:
1. The three inductive biases are chosen to map into principles studied in both neuroscience and machine learning.
2. Comprehensive ablations experiments reporting statistical significance tests
3. Authors provide a clean definition for LOC, with a detailed example of LPC calculation that helps readers in understanding the metric

weaknesses
1. Per task accuracies are not reported

---

> ### Author Rebuttal · Authors · 2025-07-30
>
> Dear Reviewer,
>
> Thank you for taking the time to review our paper and sharing your ideas for improvements with us. We are excited to see that you liked our work! To address your questions:
>
> **(1: Applicability to non-time series tasks)** Our architecture is built with timeseries data in mind, but it works with static input data as well. In this case, one may want to turn experts and routers into feedforward layers as is typically done for Mixture-of-Experts models [ref.1], but our architecture would work without any changes.
>
> **(2: Task accuracy table)** We provide some summary statistics of task accuracies across models in our answer to your following question, giving some insight into the specific effects of some of our contributions. Dropout in particular is required for certain aspects of pathway formation, but impairs accuracy, as can be seen below and in our paper, on line 242. Note that the baseline model with the highest accuracy does not use any dropout. Nonetheless, all models perform well above chance, which would be 1/16 = 6.25%. The camera-ready version of the manuscript will provide full overview tables in the appendix, with a summary overview in the main text.
>
> **(3: Ablation of parameters in A.1)** We have now run additional ablation experiments, varying the alpha parameter, dropout, router size, and embedding layer. We highlight the effect of these ablations through extracting the correlations shown in Fig.5 and Fig.6, based on 10 models run for each configuration:
>
> |  | **Mean task accuracy** | **Std. dev. task accuracy** | **Median task accuracy** | **Fig.5 correlation** | **Fig.5 p-value** | **Fig.6b correlation** | **Fig.6b p-value** |
> |---|---|---|---|---|---|---|---|
> | Baseline | 91.05%  | 8.93% | 93.26% | -0.01 | 0.6576 | -0.49 | 0.0000 |
> | Our Model ($\alpha$ = 1e-5; Router dim = 64) | 82.97% | 15.50% | 87.52% | 0.54 | 0.0000 | 0.31 | 0.0039 |
> | No dropout | 90.10% | 8.93% | 91.10% | 0.55 | 0.0000 | 0.03 | 0.7815 |
> | $\alpha$ = 1e-4 | 69.02% | 20.13% | 69.62% | -0.57 | 0.0000 | -0.37 | 0.0004 |
> | $\alpha$ = 1e-6 | 89.68% | 9.00% | 92.90% | **0.62** | 0.0000 | 0.18 | 0.1157 |
> | No task embedding | 83.04% | 14.16% | 87.05% | 0.58 | 0.0000 | **0.58** | 0.0000 |
> | Router dim = 32 | 83.15% | 14.52% | 87.15% | 0.46 | 0.0000 | 0.33 | 0.0023 |
> | Router dim = 128 | 81.95% | 16.95% | 85.98% | 0.35 | 0.0013 | 0.27 | 0.0125 |
>
> We observe that our model lies at an intersection of good accuracy and strong significant correlations for our Fig.5 and Fig.6 findings. The router dimension does not seem to strongly affect the overall pattern. As expected, the correct alpha value is critical: setting it too high leads to an over-emphasis on routing cost, causing the model to give up on learning complex tasks and fail to converge properly. Smaller alpha values increase accuracy, but form weaker pathways, as there is not enough pressure to assign tasks to the simplest possible pathways. Having no dropout naturally helps task performance, as the model now gets to use all experts at all times, but it also has the interesting effect of causing the Fig.6 correlation to disappear. This suggests that the findings of brain-like learning as shown in Fig.6 develops as an interaction of the LPC loss and the dropout. Lastly, we find that removing our task embedding layer leads to the same effects as our main model, but with an even stronger correlation. Including the embedding layer increases training speed by about 250% and was hence helpful for us to conduct our experiments. We will include these results and analysis in our final manuscript.
>
> We want to thank you again for taking the time to review our work and suggesting improvements to our manuscript. We hope our comments above could clarify all the open questions you had.
>
> References:
>
> [1] ViMoE: An Empirical Study of Designing Vision Mixture-of-Experts: https://arxiv.org/abs/2410.15732

---

> > ### Comment · Reviewer_WDiN · 2025-08-06
> >
> > I thank authors for their efforts in addressing comments. I continue to support the paper’s acceptance.

---

### Note · Authors · 2025-08-13

We would like to thank the reviewers and area chairs once again for taking the time to review our work. The added perspectives here have improved this work, and we are excited to incorporate changes into the camera-ready version such as:

1. Ablations isolating the effects of expert dropout and task embeddings
2. Discussion of cognitive effort controlled by norepinephrine release
3. Alternative difficulty metric for tasks (number of rules)
4. Per-task accuracy scores
5. Alternative measure of expert capabilities (effective rank)
6. Typos and clarifications

Our work establishes a model for studying the formation of processing pathways in the brain through a heterogeneous mixture-of-experts architecture. We are excited about the future directions that are possible to study with this modeling framework, such as how pathways are influenced by task demands beyond complexity, and the specific roles played by individual experts within a pathway. Thank you again for your time and effort.

---

### Decision · Program_Chairs · 2025-09-17

**Decision:**

Accept (poster)

**Comment:**

This paper investigates how task-specific processing pathways form in neural nets with mixture of heterogeneous experts, identifying inductive biases leading to the formation of such pathways. The paper has been generally well received by reviewers, as a novel and well-motivated conceptualization of routing in neural networks within a neuroscience perspective, which elegantly took advantage of its interdisciplinary character to present interesting results of potential broad significance for both the machine learning and neuroscience communities. Reviewers also appreciated the comprehensive ablation experiments and clear reporting of results including statistical significance tests.
Rebuttals also address some of the initial concerns of reviewers, such as the extension of the work to tasks beyond time-series, reporting of accuracy metrics for individual tasks, and the need to run additional ablation experiments to assess the impact of different model parameters.
Overall, reviewers are in broad agreement that this paper is ready for acceptance.